# Metagenomic surveillance uncovers diverse and novel viral taxa in febrile patients from Nigeria

Effective infectious disease surveillance in high-risk regions is critical for clinical care and pandemic preemption; however, few clinical diagnostics are available for the wide range of potential human pathogens. Here, we conduct unbiased metagenomic sequencing of 593 samples from febrile Nigerian patients collected in three settings: i) population-level surveillance of individuals presenting with symptoms consistent with Lassa Fever (LF); ii) real-time investigations of outbreaks with suspected infectious etiologies; and iii) undiagnosed clinically challenging cases. We identify 13 distinct viruses, including the second and third documented cases of human blood-associated dicistrovirus, and a highly divergent, unclassified dicistrovirus that we name human blood-associated dicistrovirus 2. We show that pegivirus C is a common co-infection in individuals with LF and is associated with lower Lassa viral loads and favorable outcomes. We help uncover the causes of three outbreaks as yellow fever virus, monkeypox virus, and a noninfectious cause, the latter ultimately determined to be pesticide poisoning. We demonstrate that a local, Nigerian-driven metagenomics response to complex public health scenarios generates accurate, real-time differential diagnoses, yielding insights that inform policy.

Infectious diseases place a large, global burden on human health. There are hundreds of known human pathogens, which differ in their pathogenesis, epidemiology, and therapeutic vulnerabilities. Moreover, the detection of emerging pathogens has accelerated, driven by ecological, environmental, and sociodemographic factors[1] as well as increased surveillance and diagnostic testing[2]. Accurate and timely diagnosis is essential for both clinical care and mitigation of further transmission. However, clinical diagnosis remains a challenge, as many pathogens present with highly overlapping sets of non-specific symptoms (e.g., fever, swollen lymph nodes, or malaise), and the presence of one pathogen does not preclude the presence of others (bluntly phrased by John Hickam: "patients can have as many diseases as they damn well please")[3,4]. In low- and middle-income countries (LMICs), the disease burden is often the highest, but molecular diagnostics are limited. Consequently, misdiagnosis with common pathogens such as malaria or typhoid fever, or the failure to receive a diagnosis, occurs frequently in LMICs[5-8].

The rapid determination of all species in a sample through metagenomic analysis[9-11] can identify potential causal agents of febrile illness in an unbiased, high-throughput manner. Metagenomics, alongside more sensitive approaches such as virome capture sequencing[12], can thus transform diagnostic microbiology[13] and outbreak responses[14]. The development of genomics infrastructure in Africa has enabled the continent to lead in the characterization of numerous emerging SARS-CoV-2 variants[15-19] and holds promise for the genomic interrogation of endemic pathogens[20]. Because genomics remains relatively expensive and requires technical expertise to both generate and analyze the data, it cannot be readily applied to every sample, necessitating an understanding of the most valuable applications of metagenomics in real-world settings.

e-mail: katherine_siddle@brown.edu; pardis@broadinstitute.org; happic@run.edu.ng

To evaluate the utility of metagenomic sequencing for pathogen surveillance and detection, we genomically characterized viral infections in plasma samples collected for three distinct use cases over 4 years (2017–2020) in Nigeria (Fig. 1). Nigeria has multiple factors that make it a meaningful country to study the efficacy of metagenomics in infectious disease surveillance, including a high burden of infectious disease, sequencing capacity at the African Centre of Excellence for Genomics of Infectious Diseases (ACEGID), and a strong partnership between ACEGID and national public health institutions, especially the Nigerian Centre for Disease Control (NCDC). Here, we report the results of (i) a study of suspected Lassa Fever (LF) cases, where we examine Lassa virus (LASV), non-LASV viral etiologies, and cases of co-infections; (ii) rapid investigations of three outbreaks suspected of infectious etiologies; and (iii) metagenomic diagnosis of clinically challenging cases. We report the strengths and limitations of, as well as the insights derived from, sequencing technologies in each of these settings and provide suggestions on the most effective strategies to leverage metagenomics for disease diagnosis and pathogen detection.

## Results

### Metagenomics requires stringent experimental processes and bioinformatic filtering criteria to accurately detect pathogens

The scale and complexity of metagenomic sequencing data, as well as the risk of contamination or pathogen misassignment, necessitate strict experimental and computational protocols to ensure that detected microbes are truly present. We developed procedures that greatly reduce the chance of calling false positives by (i) using both negative and positive controls, (ii) identifying intersample contamination, and (iii) developing stringent bioinformatic procedures that prioritize specificity over sensitivity (Fig. 1). Because our protocols evolved over the course of the study, we outline our recommendations and the proportion of the 593 total samples sequenced via metagenomics to which each procedure was applied (Supplementary Table 1).

Experimentally, we developed procedures to both mitigate the risk of and identify potential cases of contamination occurring in the laboratory. First, we extracted plasma samples in batches alongside non-template controls (i.e., water controls) for 574 (96.8%) samples. We designed batches to minimize the cases where samples known to be positive for a particular pathogen, such as Lassa virus (LASV), were extracted or sequenced with samples known to lack the pathogen. Before synthesizing cDNA or preparing sequencing libraries, we added a negative control (i.e., RNA isolated from K562 lymphoblast cells) and a positive control (i.e., RNA from viral seed stock spiked into RNA isolated from K562 lymphoblast cells or RNA from a previously sequenced plasma sample known to contain a specific virus) for 585 (98.7%) and 509 (85.8%) samples, respectively. At this stage, we also added sample-specific RNA spike-ins using the External RNA Controls Consortium (ERCC) sequences for each of 508 (85.7%) samples, including all samples in batches of 12 or more, increasing the probability of detecting any downstream cross contamination[21]. We sequenced the majority of samples with combinatorial dual indexes (CDIs), although we used unique dual indexes (UDIs) for the one batch sequenced on the NovaSeq 6000 system (99 or 16.7% of samples) to minimize the risk of misclassification due to index hopping.

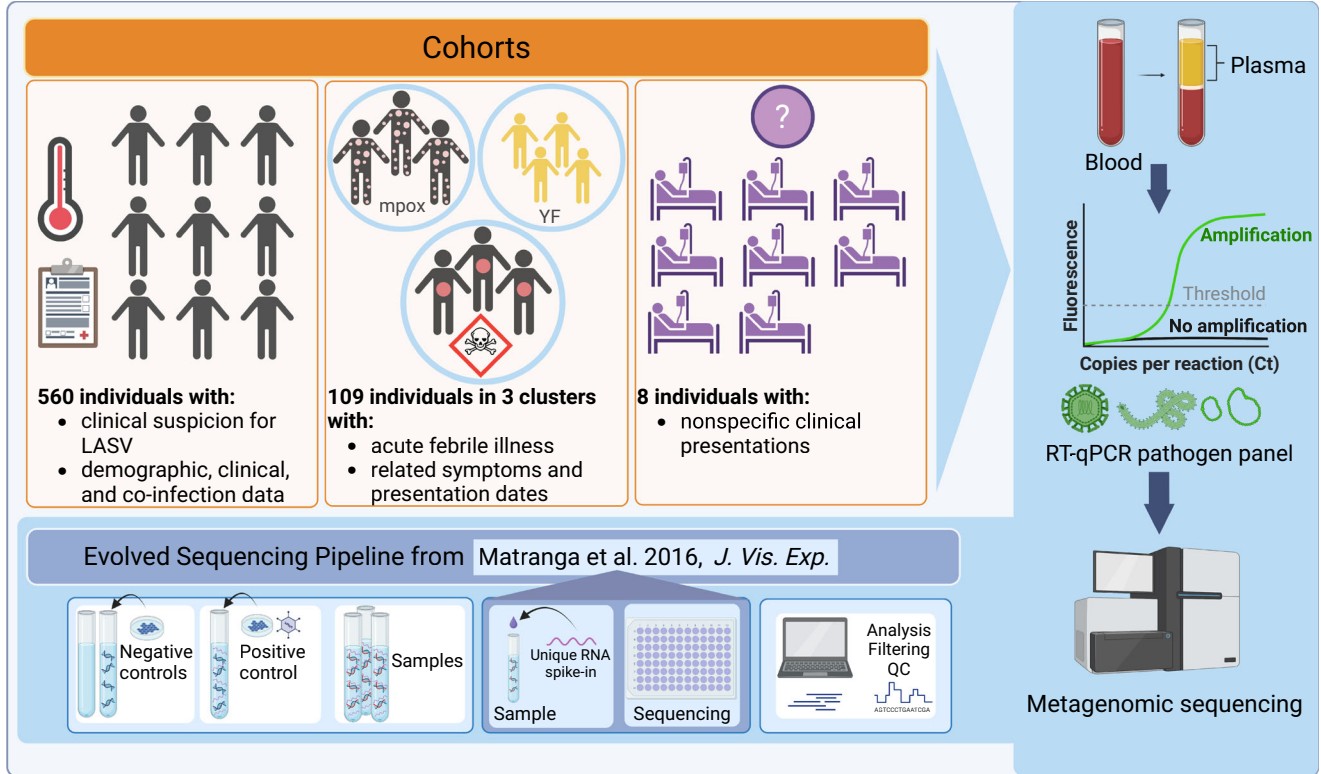

**Fig. 1 | Overview of the study design.** We conducted RT-qPCR on 670 plasma samples, followed by metagenomic sequencing of 593 of the samples, received from (i) individuals suspected to have Lassa Fever (LF; caused by Lassa virus, LASV), collected from teaching hospitals with clinical expertise in viral hemorrhagic fevers; (ii) suspected infectious disease outbreaks, collected by the Nigerian Centre for Disease Control (NCDC) and other regional clinics; and (iii) individuals with unusual or nonspecific clinical manifestations from regional clinics. We used a metagenomic pipeline inspired by Matranga et al.[21] with additional negative (i.e., water and K562 cells) and positive controls (i.e., K562 cells spiked with known viral genetic material), as well as External RNA Controls Consortium (ERCC) RNA spike-ins. We use metagenomics to identify putative causes of Lassa-like illness, to assess the role of co-infection in LASV outcomes, to determine the relationships between clinically similar acute illnesses, and to diagnose individuals with nonspecific presentations. QC, quality control. YF, yellow fever. Created with BioRender.com.

**Table 1 | Samples collected from Nigerian patients with symptoms of Lassa Fever (LF)**

| Number of Samples | LASV RT-qPCR | Hospital or Public Health Agency | State | Year | LASV Genomes | Non-LASV Genomes |
|---|---|---|---|---|---|---|
| 415 | Positive | Irrua Specialist Teaching Hospital | Edo | 2017-18 | 220 reported in Siddle et al.[14] | 37 reported in this study |
| 95 | Negative | Irrua Specialist Teaching Hospital | Edo | 2018 | 2 reported in this study | 11 reported in this study |
| 25 | Positive | Alex Ekwueme Federal University Teaching Hospital Abakaliki | Ebonyi | 2019-20 | 10 reported in this study | 0 reported in this study |
| 13 | Positive | Federal Medical Centre | Ondo | 2019 | 3 reported in this study | 0 reported in this study |
| 5 | Positive | Nigerian Centre for Disease Control | Kebbi | 2019 | 2 reported in this study | 0 reported in this study |

Amidst a 2017–2018 surge of LF in Nigeria, we generated 220 complete Lassa virus (LASV) genomes from the sequencing of 415 LASV-positive samples from the Irrua Specialist Teaching Hospital (ISTH) and reported the LASV genomes in Siddle et al.[14]. Here, we generate non-LASV genomes, as well as additional LASV genomes from 4 other cohorts.

Computationally, we chose universal, strict filtering criteria to analyze the resulting data. We first discarded samples that displayed evidence of potential cross-contamination via the ERCC spike-ins (7 of 560 samples; Supplementary Fig. 1A). We then ensured that the expected viral genomic material was identified in the positive controls via the metagenomic classification tool Microsoft Premonition[22] (Supplementary Table 2). Next, to call a virus present in a sample, we required it to have (i) at least 5 reads assigned to it by Microsoft Premonition; (ii) a greater percent of reads assigned to it than assigned to the same species in any (a) extraction-batch-specific non-template controls, (b) sequence-batch-specific positive controls, excluding the spiked in viral genomic material, and (c) sequence-batch-specific negative controls; and (iii) genome assembly of Microsoft Premonition hits with a threshold of at least 10% of the reference genome size (Supplementary Data 1, Supplementary Fig. 2). Thus, we combined a highly sensitive, but less specific, probabilistic classification tool with a highly specific, but less sensitive contig assembly step to assign pathogens to samples.

We assessed the sensitivity and specificity of our metagenomic pipeline relative to clinical RT-qPCR testing status by using data from the cohort of individuals suspected of LF. A positive Lassa virus (LASV) clinical test was defined as the amplification of either the *GPC* gene or the *L* gene via the commercially available Altona assay[23,24]. Prior clinical RT-qPCR status is an imperfect ground truth, as (i) genome degradation can occur between clinical testing and subsequent sequencing and (ii) RT-qPCR can yield false negative results for samples containing highly diverse viruses, such as LASV. Moreover, we expect PCR to be more sensitive than metagenomics due to target-specific amplification[25,26]. Nevertheless, we found that the Premonition-based thresholds yielded a sensitivity of 91.7% and a specificity of 91.6%; the additional requirement of contig assembly reduced sensitivity to 35.4% but increased specificity to minimally 96.8% (Supplementary Fig. 1B). The imperfect specificity was attributable to 3 samples that were RT-qPCR-negative but positive via sequencing. Two of these samples yielded complete, identical LASV genomes (98% and 99% complete), while the third sample yielded a partial genome. We extensively queried these samples and re-tested them via RT-qPCR (Supplementary Note, Supplementary Fig. 3), ultimately concluding that they were most likely diagnostic false negatives, a known challenge in LASV molecular detection[27,28]. In summary, our metagenomic protocols demonstrated high specificity for identifying pathogens in a given sample.

**Metagenomics identifies Lassa virus co-infections of prognostic significance as well as viral etiologies of Lassa-like illness**

We first used our metagenomic approach on 560 samples collected from population-level surveillance of individuals with symptoms consistent with LF, a viral hemorrhagic fever caused by LASV that is endemic to West African countries. We analyzed 458 RT-qPCR-positive and 95 RT-qPCR-negative samples to identify viral co-infections of prognostic significance, uncover viral etiologies of LF-like clinical syndromes in Nigeria, and characterize LASV diversity. The samples were collected between 2017 and 2020, span patients seen in 15 of 36 states and the Federal Capital Territory, and include 220 samples from which we previously reported LASV genomes[14] (Table 1).

We analyzed the metagenomics reads for other viral pathogens present in our LASV-positive samples, using the filters described above to prioritize specificity over sensitivity. We found that 7.8% (36/458) of LASV patients had a viral co-infection with at least one of the following viruses: hepatitis B, hepatovirus A, human blood-associated dicistrovirus (HuBDV), human immunodeficiency virus 1 (HIV-1), measles, parvovirus B-19, pegivirus C, and an unclassified dicistrovirus that we propose to name human blood-associated dicistrovirus 2 (HuBDV-2) (Fig. 2a). One sample was multiply co-infected with both hepatitis B and pegivirus C (Supplementary Data 1). We additionally identified viruses in 13.7% (13/95) of the RT-qPCR-negative samples, including LASV as previously discussed, as well as anellovirus, hepatitis B, HIV-1, and pegivirus C (Fig. 2a). One LASV-negative sample was multiply co-infected, with anellovirus, LASV (i.e., this sample was the PCR false negative that produced a partial genome), and pegivirus C.

Because co-infections were common among LASV-positive samples, we investigated whether they played a role in LASV outcomes. We analyzed the most frequent co-infections (i.e., pegivirus C, HIV-1, and clinically diagnosed malaria) alongside demographic information (i.e., age, sex, and pregnancy status), clinical covariates (i.e., diagnostic Ct and ribavirin treatment status), and outcomes (i.e., survived or deceased) for 400 LASV-positive individuals (Table 2). We conducted univariate logistic regression and found that diagnostic Ct value ($p < 0.001$) and receipt of ribavirin ($p = 0.01$) were significantly associated with outcomes, while age ($p = 0.06$) and co-infection with pegivirus C ($p = 0.18$) trended towards an association (Table 2, Fig. 2b–d, Supplementary Fig. 4A–E). Meanwhile, malaria co-infections, which were identified in 101 individuals, were not associated with outcomes ($p = 0.76$).

We conducted multivariate analyses with the four variables that were associated with LASV outcomes at $p < 0.25$. Prior literature suggests that these variables interact with outcomes and with one another in complex ways[29–33]. For example, Ct is a measure of the interplay between the host immune system and the virus, which may be affected by age[34] or co-infections, but Ct cannot be affected by ribavirin treatment since Ct is measured at the time of diagnosis before treatment is begun. We developed a causal directed acyclic graph[35] (DAG; Fig. 2e), informed by our univariate analyses and previous work[29–33], and conducted multivariable linear and logistic regression. Age and pegivirus co-infection were significant predictors of Ct (Fig. 2e, Table 3, Supplementary Fig. 4G); however, they were not associated with the outcome when controlling for Ct (Fig. 2e, Table 3, Supplementary Fig. 4F). We therefore concluded that the effect of age and of pegivirus co-infection status on the outcome is mediated by Ct[36]. We determined that the average causal mediation effects of age ($p = 2 \times 10^{-16}$) and of pegivirus co-infection status ($p = 0.02$) on outcome were significant via bootstrapping (Supplementary Table 3, Supplementary Fig. 4H, I).

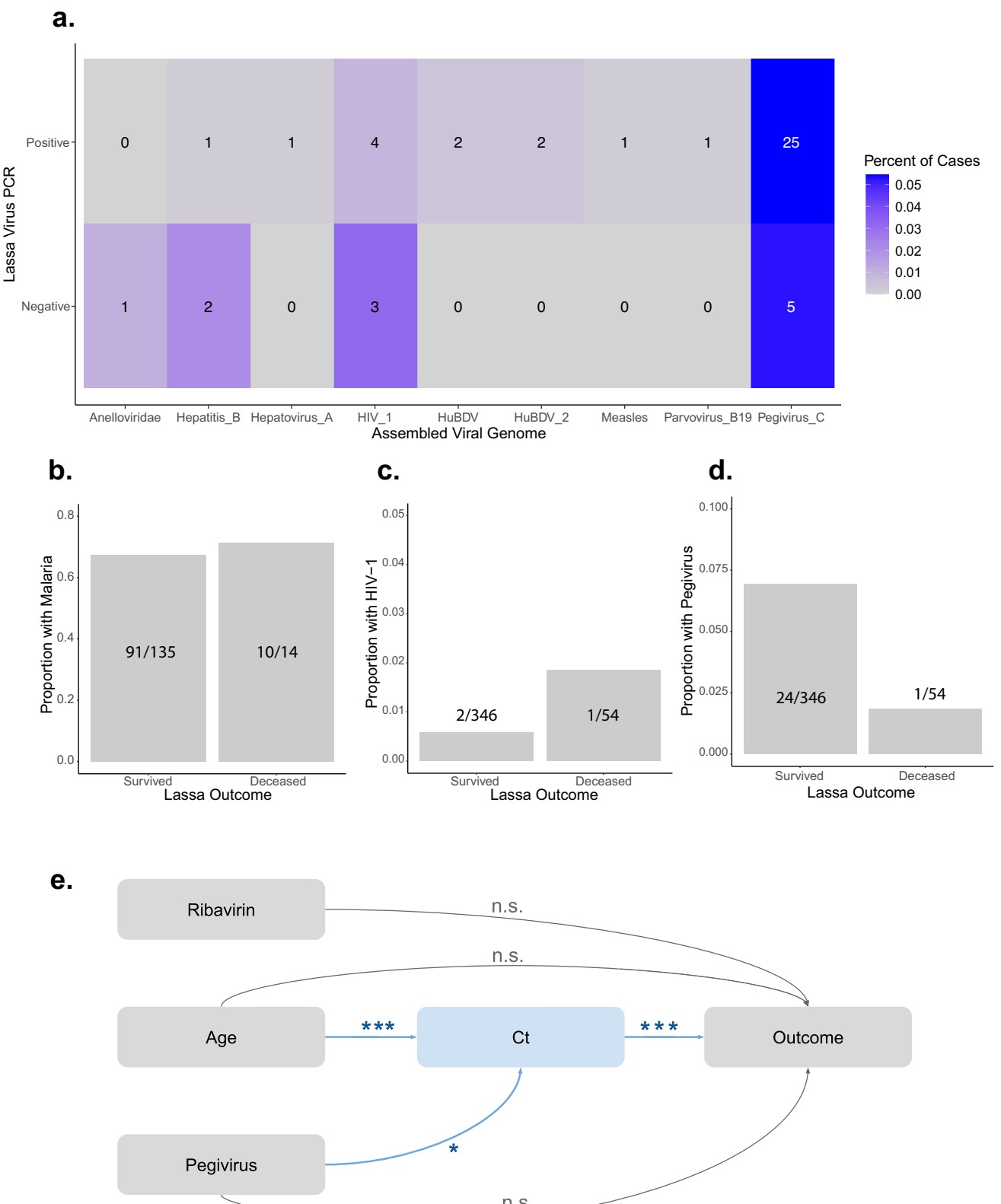

Importantly, we confirmed that there was no relationship between pegivirus C and LASV detection, i.e., due to competition for sequencing reads (Fig. 2a; Supplementary Fig. 4J). Though we cannot exclude the possibility of unknown or unmeasured confounding variables, we computed the mediational *E*-value[37], which is the risk ratio that an unmeasured confounder would need to have with both the dependent

and the independent variable to completely explain away the observed relationships. Unmeasured confounders with risk ratios of at least 1.77, 1.41, and 2.48 would be needed to fully explain the observed relationships between Ct and outcome, age and Ct, and pegivirus co-infection and Ct, respectively. In summary, our analyses suggest that older individuals have higher viral loads and thus poorer outcomes,

**Fig. 2 | Metagenomics identifies Lassa virus co-infections with prognostic implications as well as viral etiologies of Lassa-like illness. a** Metagenomics identifies Lassa virus (LASV) and non-LASV pathogens in 553 individuals presenting with symptoms of Lassa Fever (LF). Percent (color scale) and number (reported in box) of RT-qPCR-positive (458 samples) or RT-qPCR-negative (95 samples) cases containing the following non-LASV pathogens, which were each found in at least one sample: anelloviridae, hepatitis B, hepatovirus A, human immunodeficiency virus 1 (HIV_1), human blood-associated dicistrovirus (HuBDV), HuBDV-2, measles, parvovirus B19, and pegivirus C. **b–d** The proportion of surviving or deceased LASV-positive individuals who were co-infected with malaria (B), HIV-1 (**c**), or pegivirus C (**d**). **e** Causal directed acyclic graph of hypothesized relationships between ribavirin treatment, age, pegivirus C co-infection status, LASV cycle threshold (Ct) value, and outcomes. Arrows are annotated with adjusted $p$-values produced via multivariate linear (age + pegivirus → Ct; $p = 0.0007$ for age and $p = 0.023$ for pegivirus) and logistic (age + Ct + pegivirus + ribavirin → outcome; $p = 1.85 \times 10^{-12}$ for Ct) regression models. ***$p < 0.001$. *$p < 0.05$. n.s. not significant.

**Table 2 | Univariate logistic regression models identify predictors of LASV outcomes**

| Variable | No. (%) with Data | Median (IQR) or *N* (%) | Univariate *P*-value | Unadjusted OR (95% CI) |
|---|---|---|---|---|
| *Demographics* | | | | |
| Age | 380 (95.0%) | 31 (21.8–45) | 0.06 | 1.01 (1.00–1.03) |
| Sex | 398 (99.5%) | 165 (41.5%) female | 0.32 | 0.74 (0.40–1.33) |
| Pregnant | 94 (57%) | 4 (4.3%) of females | 0.36 | 3.00 (0.14–26.45) |
| *Clinical data* | | | | |
| Mean Ct | 391 (97.8%) | 36.9 (31.5–40.8) | $2.79 \times 10^{-14}$*** | 0.81 (0.76–0.85) |
| Ribavirin | 386 (96.5%) | 257 (66.6%) treated | 0.01* | 0.48 (0.27–0.87) |
| Outcome | 400 (100%) | 346 (86.5%) survived | NA | NA |
| *Co-infections* | | | | |
| Malaria | 149 (37.3%) | 101 (67.8%) | 0.76 | 1.21 (0.38–4.60) |
| HIV-1 | 400 (100%) | 3 (0.8%) | 0.34 | 3.25 (0.15–34.45) |
| Pegivirus C | 400 (100%) | 25 (6.3%) | 0.18 | 0.25 (0.01–1.24) |

No. (%), number (percent) of cases with available data. IQR interquartile range. OR (95% CI), odds ratio (95% confidence interval). CIs, ORs, and unadjusted $p$-values generated via univariate logistic regression. ***$p < 0.001$. *$p < 0.05$. NA not applicable.

while those co-infected with pegivirus C have lower viral loads and thus more favorable outcomes.

Next, we further investigated the genome sequences of several pathogens identified in the LASV-positive and LASV-negative samples, beginning with LASV itself, which is highly genetically diverse. Its distinct viral lineages segregate geographically in Nigeria[14], though most available genome sequences are from the southwestern region. Our work generated 17 new high-quality (>90% of the genome assembled) LASV genomes, 15 from PCR-positive cases and two from PCR-negative cases. We observed phylogenetic clustering of these samples by geographic origin, consistent with previous descriptions of geographic structure in LASV diversity in Nigeria (Fig. 3). Most of our genomes, including those from the PCR-negative samples, were of lineage II, and clustered according to their sampling site (Irrua in the southwestern cluster and Ebonyi in the southeastern cluster). Two genomes from samples obtained in northwestern Nigeria clustered with lineage III genomes but formed a distinct sub-clade, highlighting the extent of unsampled diversity in this poorly studied lineage.

We also more closely examined our multiple hepatitis B, HIV-1, and pegivirus C genomes. All three hepatitis B genomes, from one LASV-positive and two LASV-negative individuals, were classified as subtype E, the predominant circulating genotype in Western and Central Africa[38]. At least two of the seven HIV-1 genomes, from four LASV-positive and three LASV-negative samples, were recombinant (Supplementary Table 4). We constructed a phylogenetic tree with our 28 complete pegivirus C genomes from 23 LASV-positive and five LASV-negative individuals and the other 130 annotated sequences available in NCBI GenBank. The Nigerian genomes cluster with other African genomes, in particular those from Ghana and Cameroon, the nearest countries represented in the tree (Supplementary Fig. 5).

Finally, we report the first four Nigerian genomes of dicistroviruses, all of which were found in LASV-positive samples. Dicistroviruses have primarily been described in arthropods[39–43], though the poorly characterized human blood-associated dicistrovirus (HuBDV) was first discovered in a febrile Peruvian patient in 2018[44]. Here, we assembled the second complete HuBDV genome and another partial genome. Moreover, we assembled two additional unclassified dicistroviridae genomes, which were >96% identical to sequences produced from febrile Tanzanian children[45] and highly divergent from the HuBDV genomes (Fig. 4). We designate the clade that includes our two unclassified genomes and the three Tanzanian genomes as human blood-associated dicistrovirus 2 (HuBDV-2; Fig. 4). Our identification of unlinked cases of HuBDV and HuBDV-2 suggests that these viruses may be circulating more broadly than known in Nigeria.

## Cluster investigations yield genomic insights that inform public health interventions

Genome sequencing has successfully identified the etiologies of disease outbreaks and determined the relationships between cases within a cluster[13,46–48]. We investigated three separate outbreaks via the analysis of 109 plasma samples collected by the NCDC. We tested all samples using an RT-qPCR-based common pathogens panel (Supplementary Table 5; Supplementary Data 1) and conducted subsequent metagenomic sequencing on a subset of samples for outbreak characterization.

The first cluster investigation consisted of 71 samples collected in 2017 from patients suspected to have mpox, caused by monkeypox virus (MPXV). MPXV re-emerged in Nigeria over the same calendar year, after 40 years of absence, and sequencing of early cases suggested spillover from a local reservoir, rather than importation, as the source[49]. Here, we conducted diagnostics and sequencing from plasma samples rather than lesion swabs, which are heterogeneous samples that can be difficult to collect from those with few or no visible lesions[50]. Though plasma is a more standardized sample type, the degree to which MPXV genetic material is detectable in plasma is unknown. Of our 71 plasma samples, 35 were positive for MPXV by qPCR (Supplementary Table 6), indicating a minimum sensitivity of 49% for plasma testing (as not all patients were certain to have MPXV). We selected five MPXV-positive plasma samples—those with the highest sequencing library quantification values—for unbiased sequencing as well as hybrid capture with pan-viral target enrichment probes

(Methods). Unbiased metagenomics yielded 30 or fewer aligned read pairs for each sample, while hybrid capture yielded up to 20,000 aligned read pairs (Supplementary Fig. 6). We produced contigs capable of determining that the 5 samples belonged to the IIb clade (i.e., the clade responsible for the 2022 multinational outbreak), consistent with other outbreak reports[49]. We could not assemble complete genomes via either metagenomics or hybrid capture, likely due in part to the large genome size, reduced viral loads in the blood relative to lesions[51], and the Illumina MiSeq's sequencing capacity.

**Table 3 | Multivariate linear and logistic regression models identify predictors of LASV outcomes**

| Independent variable | P-value | Regression coefficient (β) | Standard error of β | 95% CI (β) |
|---|---|---|---|---|
| *Age + Pegivirus → Ct* | | | | |
| Age | 0.0007*** | −0.066 | 0.019 | −0.10 to (−0.02) |
| Pegivirus | 0.023* | 3.314 | 1.447 | 0.48–6.15 |
| **Independent variable** | **P-value** | **Regression coefficient (β)** | **Odds ratio (OR)** | **95% CI (OR)** |
| *Age + Pegivirus + Ct + Ribavirin → Outcome* | | | | |
| Age | 0.945 | 0.001 | 1.00 | 0.98–1.02 |
| Pegivirus | 0.758 | −0.334 | 0.72 | 0.03–4.11 |
| Ct | 1.85 × 10⁻¹²*** | −0.209 | 0.81 | 0.76–0.86 |
| Ribavirin | 0.336 | −0.364 | 0.70 | 0.33–1.47 |

*95% CI* 95% confidence interval, *OR* odds ratio, CIs, ORs, and *p*-values generated via multivariate linear and logistic regression and adjusted for covariates. ***p < 0.001. *p < 0.05. *NA* not applicable.

The second cluster investigation consisted of eight samples suspected to contain yellow fever virus (YFV), collected in 2020 from Ebonyi, Edo, and Oyo states. YFV is the etiological agent of YF and also re-emerged in Nigeria in 2017 after a 40-year absence[52]. Previously, we reported YFV in a 2018 cluster with symptoms suggestive of LF and demonstrated that the cases were more closely related to contemporary Senegalese YFV genomes than to historical Nigerian sequences[53]. After confirming YFV was found in all eight samples via RT-qPCR, we sought to characterize the genomic ancestry of the 2020 outbreak. We produced two complete YFV genomes, which belonged to the West Africa clade (Supplementary Fig. 7) and were >98% similar to sequences from the Nigerian 2018 YFV outbreak[53], suggesting cryptic transmission and persistence of the 2018 YFV strain. These data contributed to the NCDC's and World Health Organization's (WHO) efforts to accelerate vaccination campaigns and train local healthcare workers in the diagnosis and treatment of YF[54].

Finally, we received 30 samples in November 2020 from a cluster in Benue, Nigeria, that presented with headache, diarrhea, vomiting, and abdominal pain. The samples were negative for all pathogens in the RT-qPCR panel, and metagenomic sequencing of 12 samples failed to identify an infectious etiology. The NCDC ultimately expanded its differential diagnosis to include environmental causes, and the outbreak was determined to be due to pesticide poisoning[55,56]. While metagenomics of a single sample type cannot rule out an infectious cause, this investigation emphasizes that it can aid public health departments in updating their prior probabilities of specific diagnoses.

## Metagenomics identifies viral infections in undiagnosed, severe clinical cases

In the clinical setting, metagenomic sequencing offers an alternative to the enumeration of single-pathogen diagnostic tests, which can

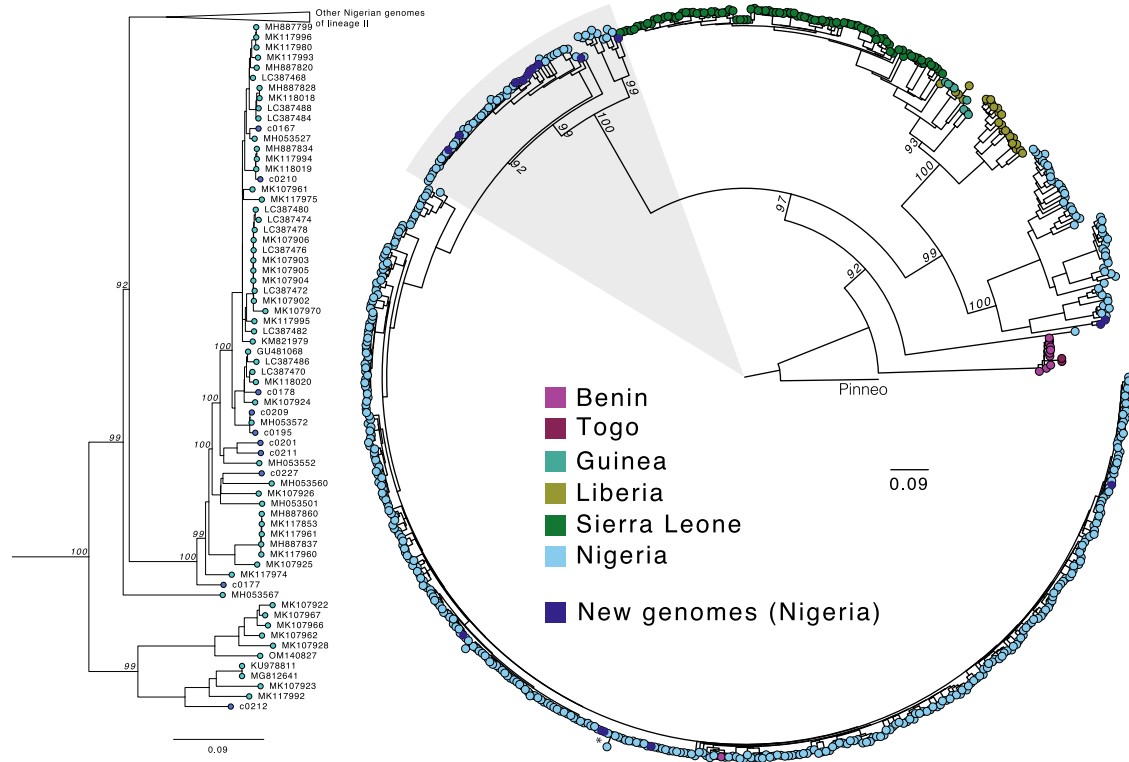

**Fig. 3 | Lassa virus genetic diversity.** Maximum likelihood phylogenetic tree of 17 new genomes (dark blue) alongside 622 published complete S segment coding sequences. Tips are colored by the country of sample origin, and the tree is rooted in the Pinneo sequence (1979). The area highlighted in gray, containing the majority of the new genomes (10/17), is shown in more detail on the left. The asterisk denotes the two RT-qPCR-negative samples that yielded complete genomes. The scale bar denotes substitutions per site. Bootstrap values are shown on key nodes.

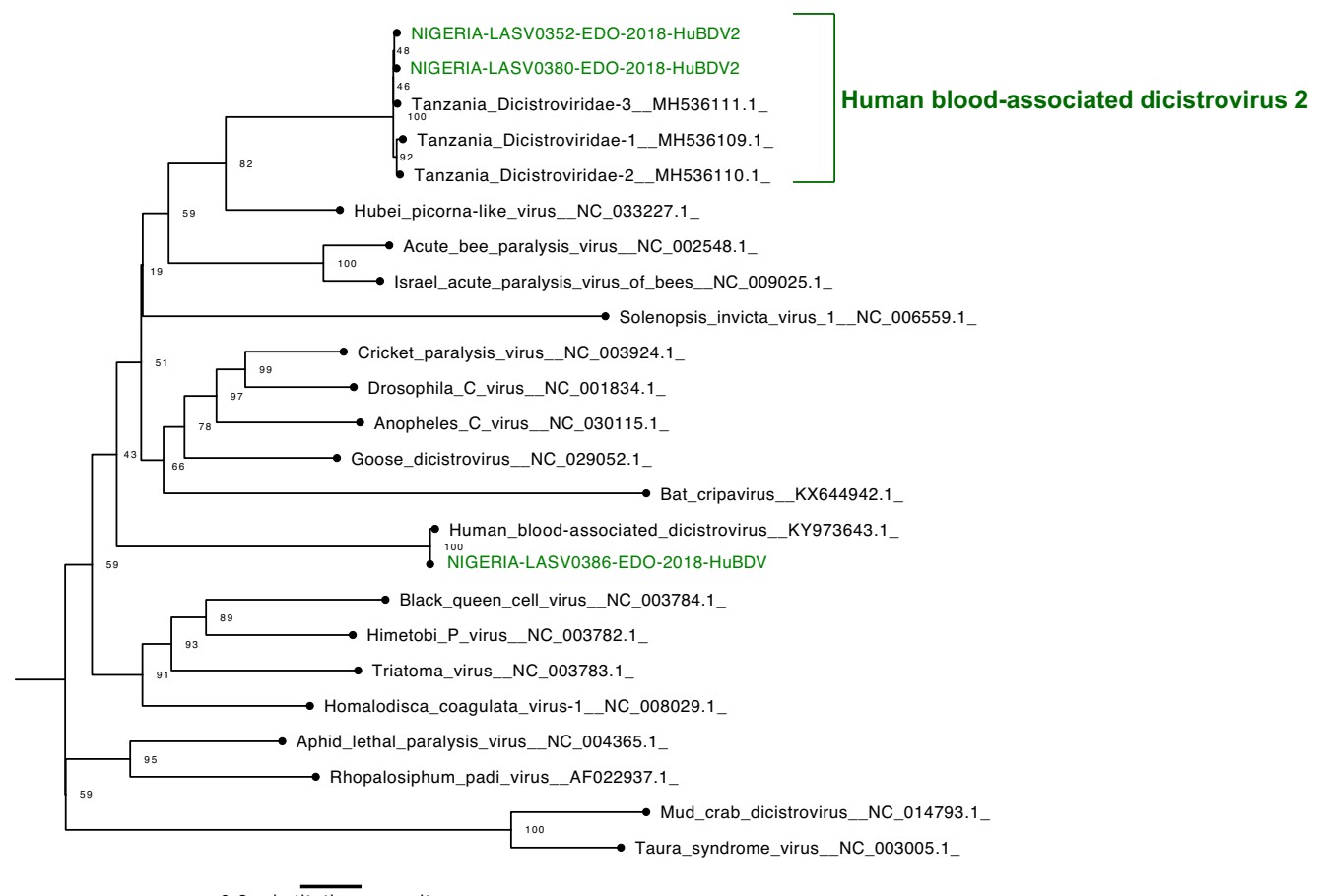

**Fig. 4 | Dicistrovirus *RdRp* (RNA-dependent RNA polymerase) genetic diversity.**
Maximum likelihood phylogenetic tree with 3 new sequences (green) alongside 21 published sequences. Generated from 2540-bp *RdRp* gene alignment. Bootstrap values for key nodes are shown. The clade that we name human blood-associated dicistrovirus 2 (HuBDV-2) is labeled.

require multiple samples and ultimately be costly and time-consuming[57]. Moreover, in Nigeria and other LMIC settings, even large hospitals currently only have the capacity to test for a small set of pathogens. We received eight plasma samples from individuals with clinical presentations consistent with an infectious etiology but without evidence of any commonly circulating pathogens, collected in 2019–2020 from Ondo, Lagos, and Ebonyi states. Clinical and demographic metrics for these cases were highly varied (Supplementary Table 7).

We first screened the eight patient samples against the RT-qPCR common pathogens panel (Supplementary Table 5; Supplementary Data 1) and failed to identify any positive hits. Via unbiased metagenomic sequencing, we identified viruses that are plausible candidates for illness in two patients. In a third sample, we detected Pegivirus C, a common infection in healthy individuals[58] that is unlikely to be the cause of the clinical syndrome. No plausible pathogenic viral taxa were detected in the remaining five samples. Here, we describe the clinical and genomic features of the cases with a putative diagnosis.

We identified reads mapping to Enterovirus B in the plasma of a child presenting with fever and seizures. We assembled a genome of Coxsackievirus-B3 (CV-B3; Fig. 5a), which is associated with both gastrointestinal illness and more serious manifestations, including myocarditis and meningitis[59,60]. The genome was most similar to a CV-B3 genome from Japan (82% pairwise sequence identity), though the VP1 gene was most closely related to a partial genome from Nigeria (88% pairwise sequence identity to GQ496547.1)[61].

We detected type IB hepatovirus A (HAV; Fig. 5b) in another child presenting with left-sided weakness, generalized lymphadenopathy, hepatosplenomegaly, and a head CT scan with evidence of a right hemispheric stroke. HAV, the causal agent of hepatitis A, is transmitted fecal-orally, typically presents with acute gastrointestinal manifestations, and rarely causes death[62]. This patient's symptoms are not consistent with the textbook presentation of hepatitis A, though cases of neurological sequelae associated with HAV have been documented[63–66]. We thus interpret the metagenomic sequencing results with caution, as it is possible that HAV is an incidental finding. However, we only identified HAV in 1 of our 592 other samples, suggesting that it is an uncommon co-infection and lending support to the possibility that this patient presented with an unusual manifestation of HAV.

## Discussion

Here, we describe a highly specific metagenomic sequencing protocol, which we use to investigate viral etiologies of fever in Nigeria in three contexts (Fig. 1). Nigeria's high infectious disease burden, including endemic (e.g., malaria), emerging (e.g., Dengue virus) and re-emerging (e.g., LASV) pathogens, advanced sequencing capacity, and robust public health system make it a compelling place to study the role of metagenomics in infectious disease surveillance.

Our genomic investigations uncovered 13 distinct viruses using a single pipeline, informing public and patient health. Our MPXV investigation demonstrated the benefit of targeted approaches (e.g., qPCR and hybrid capture) when a pathogen is suspected while also

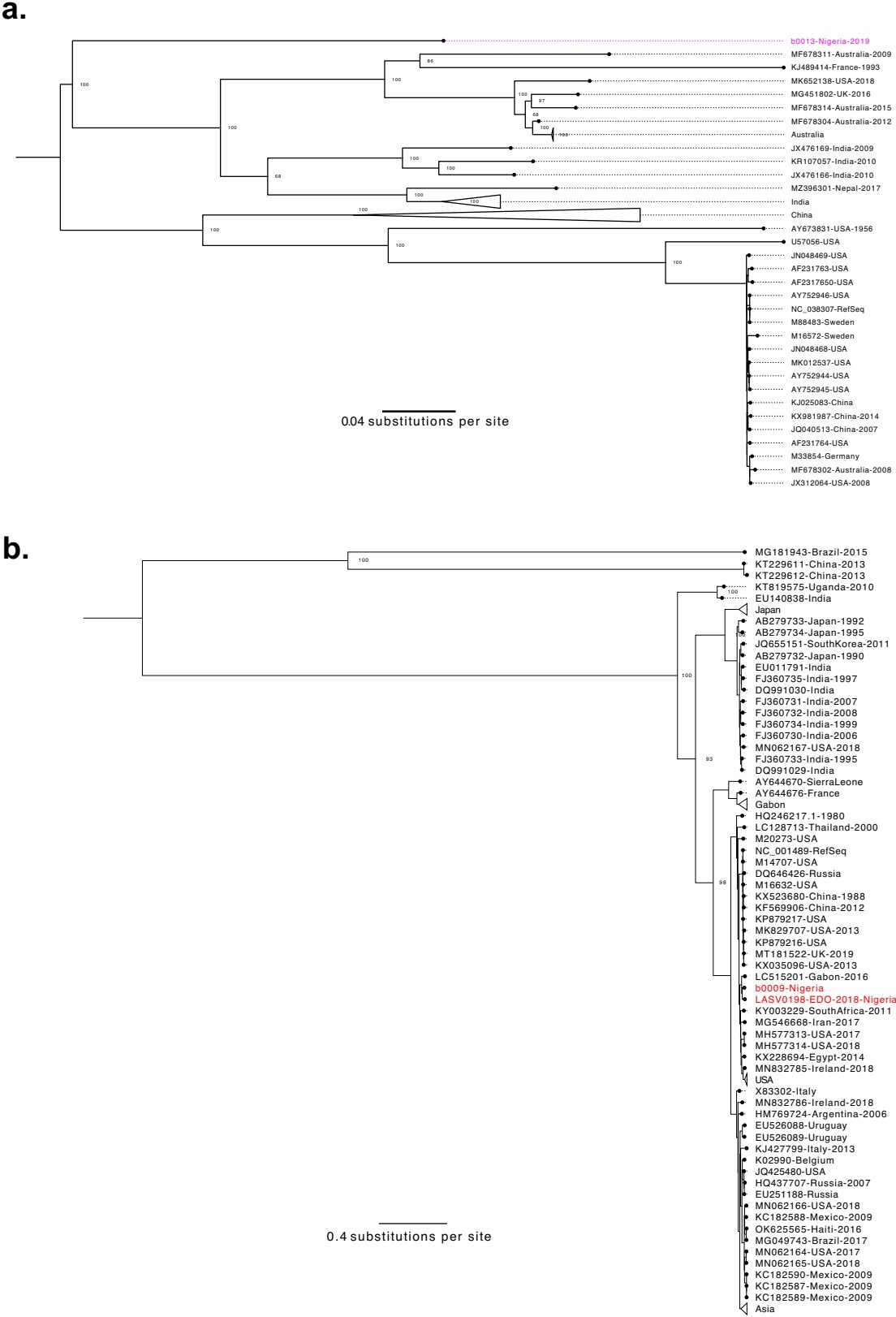

**Fig. 5 | The genetic diversity of pathogens identified in undiagnosed, severe clinical cases. a** Coxsackievirus B3 (CV-B3) genetic diversity. Maximum likelihood phylogenetic tree with one new sequence (pink) alongside 63 full-length, published sequences. Generated from whole-genome alignment (7447 bp). Bootstrap values for key nodes are shown. **b** Hepatovirus A genetic diversity. Maximum likelihood phylogenetic tree with two new sequences (red) alongside 105 full-length, published sequences. Generated from whole-genome alignment (7736 bp). Bootstrap values for key nodes are shown.

demonstrating that MPXV can be detected and subtyped from plasma samples. Additionally, we identified the poorly described HuBDVs, highlighting the need for further research while emphasizing the importance of metagenomics in detecting uncommon pathogens. Indeed, the identification of unlinked Nigerian cases of HuBDV-1 and HuBDV-2, previously identified solely in Peru and Tanzania, respectively, suggests that human dicistrovirus infections may be more widespread than previously suspected. Meanwhile, HIV and hepatitis B are major causes of morbidity and mortality, both globally and in Nigeria[67–70], and were found in multiple individuals in our LASV-negative cohort, representing worthy candidates for follow-up testing of patients with symptoms of LF. We also uncovered the possible protective effect of pegivirus C in LASV infection. Over 5% of the LASV-positive cohort was co-infected with pegivirus C, which is consistent with its estimated prevalence of 7–12% in healthy West African blood donors[71,72]. Our causal mediation analysis suggests that pegivirus C contributes to beneficial LASV outcomes via the mediation of LASV viral load. While this finding is consistent with favorable prognostic reports from hepatitis C, HIV, and Ebola virus patients co-infected with pegivirus C[32,33,58], we emphasize the need for further epidemiological and mechanistic research.

Our phylogenetic reconstructions also produced actionable public health insights. Our finding that 2020 YFV cases were descendants of 2018 Nigerian cases indicated the presence of cryptic transmission and prompted the NCDC and the National Primary Health Care Development Authority (NHPCDA) to accelerate their vaccination efforts. On the other hand, our pegivirus C genomes were interspersed with those from Cameroon, Sierra Leone, and Uganda, emphasizing that transmission patterns in Nigeria are a result of both importation and internal circulation. Finally, we identified undersampled viral diversity, both by sequencing LASV samples from Kebbi, which form a clade within lineage III, and by generating the first complete coxsackievirus B3 genome on the African continent.

Metagenomic sequencing is a powerful diagnostic platform but requires careful analysis and interpretation. By using multiple experimental controls followed by strict computational thresholds, we achieved a high specificity for pathogen identification. Nevertheless, the molecular detection of a pathogen does not establish causality nor fulfill Koch's postulates[73]. This is particularly important in individual cases, where one cannot rely on statistical enrichment (e.g., case–control comparisons) or pseudo-replication (e.g., cluster investigations). For example, we found HAV in a child lacking the traditional hepatitis A presentation, expanding rather than narrowing the differential. Moreover, we failed to identify a pathogen in some samples. While some cases were truly negative for an infectious etiology, such as those from individuals with pesticide poisoning[55,56], metagenomic sensitivity is limited by biological and technical factors. Some pathogens are undetectable in specific tissue compartments or disease stages. Additionally, technical challenges limit the sensitivity of metagenomic (vs. amplicon-based) sequencing for certain pathogens or sample types[12], particularly with certain technologies (e.g., lower-throughput sequencing machines, which are more widely available in LMICs). Finally, non-viral pathogens, which we do not consider here[28], require exploration to fully eliminate microbial etiologies.

Practical barriers currently prohibit the widespread use of metagenomics for diagnosis, making it most suitable as a complement and necessary prerequisite to the development of molecular assays. Routine surveillance of undiagnosed cases through metagenomics can highlight the pathogens to prioritize for diagnostic capacity building in a given cohort. We found metagenomics to be particularly valuable in cluster investigations, where multiple instances of detection increase diagnostic certainty, and the resulting genomic data enables the study of transmission and development of policy measures[74,75]. For hospitalized patients needing a diagnosis, we advocate for a tiered approach, where point-of-care, multiplexed diagnostics are made

available in clinical settings while academic and public health partnerships are established so that negative samples can be rapidly investigated via unbiased sequencing.

Here, we offer real-time insights into the etiologies of febrile illness and the genetic diversity of circulating pathogens in Nigeria. As we move beyond the SARS-CoV-2 pandemic, the genomic infrastructure established in LMICs[20] presents an unprecedented opportunity to use infectious disease genomics in a thoughtful manner to maximize the benefit to human health.

## Methods
### Patient recruitment and ethics statement
We obtained samples through studies reviewed and approved by institutional review boards (IRB) at multiple sites, including Irrua Specialist Teaching Hospital (Nigeria), Redeemer's University (Nigeria), Harvard University (Cambridge, Massachusetts), and the National Health Research Ethics Committee (Nigeria). The specific cohorts covered by each IRB are described below.

Institutional review boards of ISTH (Irrua, Nigeria), Redeemer's University, and Harvard University (Cambridge, Massachusetts) assessed and approved the study before the start of research activities. De-identified clinical samples and demographic and clinical data were collected under (i) a waiver of consent, approved by the ISTH Research Ethics Committee, or (ii) under the written informed consent of participants for participation in a separate study that analyzed human genetic material. The waiver of consent enables the analysis of pathogen genomic data and de-identified demographic and clinical data but not the analysis of human genetic material. For the purposes of the work in this manuscript, the sample sets are equivalent in terms of data availability.

ISTH is a federal teaching hospital and LF specialist center located in an area of high LASV endemicity. ISTH treats hundreds of LF patients each year and tests thousands of patient samples for LASV, including from patients presenting to ISTH and from samples sent by doctors elsewhere in Nigeria. Because ISTH is a National Centre of Excellence for LF management, suspected patients are referred to the hospital for management from both private practices and surrounding hospitals and clinics. Among patients presenting to ISTH, LF was considered as a possible cause of undiagnosed acute febrile illness in patients with (a) fever ≥38 °C and no improvement after 2 days of antimalarials or antibiotics, or (b) fever ≥38 °C with at least one LF-associated symptom: bleeding from mucosal surfaces or injection sites, deafness, conjunctivitis, facial edema, hypotension, spontaneous abortion, seizures, encephalopathy, or acute kidney injury. Plasma was isolated from a venous blood draw collected from all suspected cases for diagnostic testing.

In addition to samples collected at ISTH, ACEGID at Redeemer's University received samples from multiple sites suspected of an infectious etiology. Samples in clinical excess (e.g., samples from individuals with critical, undiagnosed conditions) from Federal Teaching Hospital Abakaliki (FETHA) and Federal Medical Center (FMC) Owo were received via a study approved by the National Health Research Ethics Committee (NHREC, Nigeria) under a waiver of consent. Samples from individuals in case clusters were received from the Nigerian Centre for Disease Control (NCDC). As a regulatory body for public health in Nigeria, NCDC collects samples, some of which are sent to ACEGID for rapid sequencing in the context of public health emergencies. All samples received contained plasma isolated from venous blood draws.

### RNA extraction and screening by qPCR
Prior to RT-qPCR testing, suspected LASV samples were inactivated with Buffer AVL (Qiagen), and RNA was extracted using the QIAmp Viral Mini extraction kit (Qiagen). At ISTH, patients meeting the criteria for suspected LASV were tested using 2 RT-qPCR assays, one targeting

the *GPC* gene (RealStar LASV RT-PCR Kit 1.0 CE, Altona Diagnostics, Hamburg, Germany) and a second targeting the LASV L segment[23,27]. At Redeemer's University, samples suspected of LASV infection were tested using either the RealStar® Lassa Virus RT-PCR Kit 2.0 targeting the *L* and *GPC* genes in one assay or an in-house assay adopted from Nikisins et al. [27]. Samples not suspected of LASV virus were tested for YFV, Chikungunya virus (CHKV), West Nile Virus (WNV), Zika virus (ZIKV), O'nyong-nyong virus (ONNV), Ebola virus (EBOV), Dengue, flaviviruses, and alphaviruses using an RT-qPCR common pathogens panel. Primers were adopted from previous work (Supplementary Table 5)[76–80].

Suspected MPXV samples underwent DNA extraction using the Qiagen DNeasy kit and were tested via qPCR using previously published primers[81].

Samples that were negative for LASV when tested at ISTH but which assembled a partial or complete LASV genome were re-tested at the Broad Institute using a previously published primer set[27].

## Metagenomic sequencing

Unbiased metagenomic sequencing was performed from extracted nucleic acids as previously described[14]. Briefly, we used TurboDNase treatment to remove DNA from all samples except those positive for MPXV by qPCR. We synthesized double-stranded cDNA using random hexamer priming. Sequencing libraries were constructed using the Nextera XT library preparation kit (Illumina) and sequenced on an Illumina instrument with 100-bp, paired-end sequencing. For MPXV samples, we additionally performed targeted enrichment with a pan-viral probe set targeting 356 viral species as previously described[82]. Samples were prepared and sequenced at either the Broad Institute or Redeemer's University. Metagenomic sequencing data from LASV-positive cases collected from ISTH were previously reported[14], but the non-LASV reads were not analyzed.

RNA-based controls, including commercially purchased RNA from K562 cells (negative control) and RNA from K562 cells, spiked with Ebola virus RNA (Makona variant; positive control), were added prior to cDNA synthesis. For one batch each, we used extracted RNA from a previously sequenced sample known to contain LASV or mumps virus as a positive control (Supplementary Tables 1 and 2).

## Genomic data analysis

Samples with fewer than 1000 total reads were discarded. We also discarded samples with low ERCC spike-in purities, defined as the number of reads assigned to the major ERCC spike-in divided by the total number of reads assigned to any ERCC spike-in. For each ERCC spike-in, we determined the mean and standard deviation of its purity scores across samples and batches. Samples with greater than 100 total reads assigned to any ERCC spike-in, for which purity was both <99% and less than three standard deviations below the mean for that spike-in, were discarded as previously described[83].

We then analyzed the sequencing reads using the Microsoft Premonition metagenomics pipeline[22] (https://microsoft.com/premonition) with default settings to assign reads to viral taxa. This pipeline uses an alignment-based approach (e.g., using k-mers) to map sequences against a large reference database, rather than filtering out human reads a priori, coupled with a statistical model to assign probabilities to the assignment of individual reads to taxonomic levels. Access to the pipeline is via a web interface, with cloud-based processing of sequence datasets on the Microsoft Azure platform, allowing rapid generation and retrieval of results. Viral hits were filtered to remove those with less than five reads. Samples were required to have a greater percentage of reads assigned to a particular virus than the percentage of reads assigned to that virus across all batch-specific controls. We attempted to assemble complete genomes for all remaining viral hits. For genome assembly, we used the viral-ngs pipeline[84] (version v2.1.8; https://github.com/broadinstitute/viral-ngs).

For most viruses, we performed reference-based assembly using the RefSeq genome of each virus (Supplementary Data 1). We performed de novo assembly with reference-genome-guided refinement[84] for the following genetically diverse viruses: LASV, Enterovirus B, HIV-1 (e.g., for samples lacking a sufficiently similar reference genome for reference-based assembly), and pegivirus C. Hits that assembled a genome of at least 10% of the reference genome length were retained for downstream analysis. For segmented viruses, we required 10% of the full genome length (i.e., the sum of individual segment lengths) to be assembled. Bacterial and eukaryotic taxa were not considered.

We noticed that >50% of samples with reads mapped to Pegivirus A also had reads mapped to Pegivirus C. In all such cases, we could not assemble a Pegivirus A genome; for the majority of the samples, we assembled a Pegivirus C genome. Therefore, we attempted the assembly of both Pegivirus A and Pegivirus C for all samples meeting the reads-based thresholds for Pegivirus A, regardless of any reads mapping to Pegivirus C. We only assembled Pegivirus C genomes across all cases. This highlights a fundamental challenge of metagenomic classification—that highly related species can be misclassified—but provides support for our combined approach.

Finally, we manually filtered the results to remove known contaminants (e.g., the reverse transcriptase of murine leukemia virus) and to group distinct taxa that were identified within the same family. Specific torque teno viruses were grouped with the anelloviridae family, and the unclassified Tanzanian dicistroviridae sequences were grouped together with our highly related, unclassified dicistroviridae sequences and designated HuBDV-2.

## Causal mediation analysis

Outcomes data and associated clinical covariates were collected at ISTH and de-identified by clinicians. Missing data were not imputed, though, for cases with missing pregnancy data, we assumed that females <10 years old and >60 years old were not pregnant. We also assumed that individuals who were not admitted to the hospital survived and additionally did not receive IV-administered ribavirin. The analyzed Ct values were the average of the Ct values for the L segment and S segment for samples tested via a multi-target RT-qPCR test. If only one Ct value existed, either due to failed amplification of one target or the use of a single-target RT-qPCR test, the single Ct value was used instead. We assessed the relationship between each variable and LASV outcome using univariate logistic regression, generating p-values and unadjusted odds ratios (Table 2). We decided a priori that any variable associated with the outcome at $p < 0.25$ in the univariate analysis would be included in the multivariate logistic regression models.

We fit the linear and logistic regression models (Table 3) to our data using the stats package (version 4.1.1) in R (version 4.1.1). The causal mediation analyses were performed using the Baron & Kenny framework[36] and the mediation package (version 4.5.0; Supplementary Table 3). Mediational *E*-values were calculated using the website created by Mathur et al.[85] with the contrast of interest in the exposure set as 1 for pegivirus co-infection status and 10 years for age.

## Viral genotyping

Viral subtyping was carried out using several pathogen-specific tools with default settings: Hepatitis A (https://www.rivm.nl/mpf/typingtool/hav/), Enterovirus B (https://www.rivm.nl/mpf/typingtool/enterovirus/), HIV (Stanford University HIV Drug Resistance Database; https://hivdb.stanford.edu/hivdb/), and Hepatitis B (https://www.genomedetective.com/app/typingtool/hbv/).

## Phylogenetic reconstruction

We constructed maximum likelihood phylogenetic trees for multiple pathogens. For LASV, we used all sequences with greater than 90% unambiguous length generated in this work. We downloaded from NCBI GenBank all available S segment sequences (June 23, 2022).

Sequences were filtered to retain only those sequences with complete coding sequences (CDS) from either *H. sapiens* or *M. natalensis* hosts. Due to the poor coverage of the region between the *GPC* and *NP* CDS regions, we extracted and concatenated the two CDS from the S segment for subsequent analysis and performed a multiple sequence alignment of the concatenated sequences using MAFFT[86]. We estimated a maximum-likelihood phylogeny with IQ-TREE v2.0.3[87,88] using a general time reversible nucleotide-substitution model with a gamma distribution of rate variation among sites and 1000 iterations of ultrafast bootstrapping. We rooted the tree on the Pinneo sequence (1979).

For the dicistroviruses, we downloaded from NCBI GenBank 21 sequences from multiple species, which we aligned with our 3 study sequences using MAFFT[86]. The *RdRp* gene was extracted using Geneious Prime v2023.0.4 (www.geneious.com). We estimated a maximum-likelihood phylogeny with IQ-TREE v1.6.12[89] with a TVM + F + G4 nucleotide-substitution model and ultrafast bootstrapping[90,91].

For pegivirus C, we downloaded from NCBI GenBank all available full-length, properly annotated sequences (February 28, 2023; 130 sequences), which we aligned with our 28 study sequences from individuals suspected of LF using MAFFT[86]. We estimated a maximum-likelihood phylogeny with IQ-TREE v1.6.12[89] with a GTR + F + I + G4 nucleotide-substitution model and ultrafast bootstrapping[90,91].

For YFV, we downloaded from the YFV Phylogenetic Typing Tool[92] representative full-length sequences from African countries, which we aligned with our 2 study sequences using MAFFT[86]. We estimated a midpoint-rooted maximum-likelihood phylogeny with IQ-TREE v1.6.12[89] with a GTR + F + I nucleotide-substitution model and ultrafast bootstrapping[90,91].

For coxsackievirus-B3, we downloaded from NCBI GenBank all available full-length sequences (March 26, 2023; 63 sequences), which we aligned with our study sequence using MAFFT[86]. We estimated a maximum-likelihood phylogeny with IQ-TREE v1.6.12[89] with a GTR + F + I + G4 nucleotide-substitution model and ultrafast bootstrapping[90,91].

For hepatovirus A, we downloaded from NCBI GenBank all available full-length sequences (March 26, 2023; 105 sequences), which we aligned with our 2 study sequences using MAFFT[86]. We estimated a maximum-likelihood phylogeny with IQ-TREE v1.6.12[89] with a GTR + F + I + G4 nucleotide-substitution model and ultrafast bootstrapping[90,91].

Trees were visualized with FigTree v1.4.4[93] and are midpoint-rooted unless otherwise specified. Large clades with no Nigerian or study sequences were collapsed and labeled with location information.

### Reporting summary

Further information on research design is available in the Nature Portfolio Reporting Summary linked to this article.

## Data availability

The raw reads, and complete pathogen genomes generated in this study have been deposited in the Sequence Read Archive (SRA) and NCBI GenBank, respectively, under BioProject accession codes PRJNA824010 and PRJNA436552. Sample metadata (collection date, state, age, sequencing machine, sequencing batch, etc.), metagenomic read classification data for all samples and controls, viral genome assembly data, reference sequence accession numbers, and RT-qPCR results generated in this study are provided in the Supplementary Data 1 file.

## Code availability

Open source software used in this study is available at https://github.com/broadinstitute/viral-ngs[84] (i.e., pipelines for viral genomic analyses; v2.1.8) and at https://github.com/bpetros95/lassa-metagenomics[94] (i.e., code for statistical analyses; developed in R v4.1.1 with packages bda

v15.2.5, mediation v4.5.0, ROCR v1.0–11, stats v4.1.1, and tidyverse v2.0.0). Information about the Microsoft Premonition metagenomics pipeline is available at https://microsoft.com/premonition. Individuals can access the pipeline ahead of its public release by clicking the "Contact us for availability" button and mentioning this work or by emailing Simon Frost at Frost.Simon@microsoft.com.

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

## Acknowledgements

Access to the Microsoft Premonition metagenomics pipeline was made freely available for this study for research purposes only. This work was supported by grants from the National Institute of Allergy and Infectious Diseases (U54HG007480 and U01AI151812 to C.T.H. and P.C.S., U19-AI110818 to P.C.S.), the World Bank (ACE019 and ACE-IMPACT to C.T.H.), and the National Institute of General Medical Sciences (T32GM007753 and T32GM144273 to B.A.P.). P.C.S. is an investigator supported by the Howard Hughes Medical Institute (HHMI). This work is made possible by support from Flu Lab and a cohort of generous donors through TED's Audacious Project, including the ELMA Foundation, MacKenzie Scott, the Skoll Foundation, and Open Philanthropy. The content is solely the responsibility of the authors and does not necessarily represent the official views of the National Institutes of Health.

## Author contributions

Conceptualization: K.J.S., P.C.S., and C.T.H. Methodology: J.U.O., B.A.P., K.J.S., P.E.E., O.A., S.B.M., P.D.I., A.P., P.N., C.T., J.Q., and S.F.S. Software: S.D.W.F., E.K.J., A.P., D.P., C.T., K.J.S., B.A.P., P.E.O., J.U.O., and P.N. Formal analysis: J.U.O., B.A.P., P.E.O., S.B.M., K.J.S., P.N., and C.T. Investigation: J.U.O., B.A.P., P.E.E., O.A., P.N., S.B.M., P.D.I., I.O., A.P., O.S.G., J.O.A., E.A.U., A.P.E., O.B., M.A., P.N., C.T., J.Q., L.S., N.O., N.A.A., K.O., O.O., C.A., N.A., O.A., S.O., P.O.O., O.A.F., I.K., C.I., K.J.S., P.C.S., and C.T.H. Data curation: J.U.O., P.E.O., B.A.P., K.J.S., and S.B.M. Resources: I.O., J.O.A., E.A.U., A.P.E., O.B., M.A., L.S., N.O., N.A.A., K.O., O.O., C.A., N.A., O.A., S.O., P.O.O., C.I., C.T.H., D.P., and A.A.L. Writing—original draft: J.U.O., B.A.P., K.J.S., and P.E.O. Writing—review and editing: all authors; Visualization: B.A.P., K.J.S., J.U.O., and P.E.O. Supervision: O.A.F., I.K., K.J.S., P.C.S., and C.T.H. Funding: P.C.S. and C.T.H.

## Competing interests

P.C.S. is a co-founder and shareholder of Sherlock Biosciences and Delve Bio, a Board member and shareholder of Danaher Corporation, and has filed IP related to genomic sequencing and diagnostic technologies. S.D.W.F., E.K.J., and A.P. are employees of Microsoft Corporation. S.D.W.F. is a co-founder of DiosSynVax Ltd. and has filed IP relating to antiviral vaccine technologies, including candidates for Lassa virus. The remaining authors declare no competing interests.

## Additional information

Judith U. Oguzie [1,2,19], Brittany A. Petros [3,4,5,6,19], Paul E. Oluniyi [1,2,7,19], Samar B. Mehta[8], Philomena E. Eromon[2], Parvathy Nair[9], Opeoluwa Adewale-Fasoro[1,2], Peace Damilola Ifoga[1,2], Ikponmwosa Odia[10], Andrzej Pastusiak[11], Otitoola Shobi Gbemisola[2], John Oke Aiyepada[10], Eghosasere Anthonia Uyigue[10], Akhilomen Patience Edamhande[10], Osiemi Blessing[10], Michael Airende[10], Christopher Tomkins-Tinch [3,12], James Qu[3], Liam Stenson[3], Stephen F. Schaffner [3], Nicholas Oyejide[2], Nnenna A. Ajayi [13], Kingsley Ojide[13], Onwe Ogah[13], Chukwuyem Abejegah[14], Nelson Adedosu[14], Oluwafemi Ayodeji[14], Ahmed A. Liasu[14], Sylvanus Okogbenin[10], Peter O. Okokhere[10], Daniel J. Park [3], Onikepe A. Folarin [1,2], Isaac Komolafe[1,2], Chikwe Ihekweazu[15], Simon D. W. Frost[11,16], Ethan K. Jackson[11], Katherine J. Siddle [3,17,20] ✉, Pardis C. Sabeti [3,9,12,18,20] ✉ & Christian T. Happi [1,2,10,18,20] ✉

[1]Department of Biological Sciences, Faculty of Natural Sciences, Redeemer's University, Ede, Osun State, Nigeria. [2]African Centre of Excellence for Genomics of Infectious Diseases (ACEGID), Redeemer's University, Ede, Osun State, Nigeria. [3]Broad Institute of Harvard and MIT, Cambridge, MA, USA. [4]Harvard-MIT Program in Health Sciences and Technology, Cambridge, MA 02139, USA. [5]Harvard/MIT MD-PhD Program, Boston, MA 02115, USA. [6]Systems, Synthetic, and Quantitative Biology PhD Program, Department of Systems Biology, Harvard Medical School, Boston, MA 02115, USA. [7]Chan Zuckerberg Biohub, San Francisco, CA, USA. [8]Department of Medicine, University of Maryland Medical Center, Baltimore, MA, USA. [9]Howard Hughes Medical Institute, Chevy Chase, MD, USA. [10]Irrua Specialist Teaching Hospital, Irrua, Edo State, Nigeria. [11]Microsoft Premonition, Redmond, WA, USA. [12]Department of Organismic and Evolutionary Biology, Harvard University, Cambridge, MA, USA. [13]Alex Ekwueme Federal University Teaching Hospital, Abakaliki, Nigeria. [14]Federal Medical Center, Owo, Ondo State, Nigeria. [15]Nigeria Center for Disease Control, Abuja, Nigeria. [16]London School of Hygiene and Tropical Medicine, London, UK. [17]Department of Molecular Microbiology and Immunology, Brown University, Providence, RI, USA. [18]Department of Immunology and Infectious Diseases, Harvard T.H. Chan School of Public Health, Harvard University, Boston, MA, USA. [19]These authors contributed equally: Judith U. Oguzie, Brittany A. Petros, Paul E. Oluniyi. [20]These authors jointly supervised this work: Katherine J. Siddle, Pardis C. Sabeti, Christian T. Happi. ✉e-mail: katherine_siddle@brown.edu; pardis@broadinstitute.org; happic@run.edu.ng

