## [Peer Review File · Nature Communications]

Metagenomic surveillance uncovers diverse and novel viral taxa in febrile patients from NigeriaREVIEWER COMMENTS

Reviewer #1 (Remarks to the Author):

This work addresses a critical need for improved pan-pathogen surveillance in public health and clinical settings, and describes results using the most common method currently applied to this problem: pure/unbiased metagenomics. This is an expensive and laborious undertaking in any healthcare setting, and it is to the credit of the Nigerian authors that they were able to perform it in multiple contexts and on a large number of samples, albeit >200 of which have been previously reported. The work is potentially impactful and includes some novel findings - notably of a discovery of a new dicistrovirus and high prevalence of pegivirus C. However, there are a number of critical shortcomings in the methods, and some confusing reporting that should be addressed.

General comments:

The findings are interesting but, having grouped together several cohorts and experiments, the authors have created a patchwork analysis that rests entirely on the strength of their methods. Unfortunately, the methods are either insufficiently thought out, or insufficiently described, and the experiments are missing essential process controls for contamination.

As is normal for metagenomics, the sequence data are noisy, with a lot of potential false positives among PCR-negative samples (eg Fig 2B), and presumably vice versa. This requires careful reporting. Finding true positives missed by PCR is an admirable goal - and I agree this is plausible particularly for diverse viruses - but it requires unequivocal demonstration that the metagenomics method delivers true positives and not a collection of spurious reads that are either contamination, index misassignment (aka index hopping), or misclassification. Figure 2B and much of the rest of the paper do not convince me that the authors have selected a meaningful lower limit of detection (5 reads), or that they have adequately minimised cross-detection (eg with unique dual indexes) or accounted for possible contamination (water controls are not enough).

I also question the authors' decision to conflate metagenomics with targeted capture; the latter is an excellent method and clearly was beneficial here for the MPXV samples, but it is distinct from unbiased metagenomics. Conflating the two looks like an attempt to elevate the success rate of unbiased metagenomics. Since this method only relates to the MPXV cohort, this adds further confusion to what is already quite a complex report. The authors should consider focusing specifically on unbiased metagenomics, as this is the bulk of the work, and expand the methods and results substantially to demonstrate sensitivity and specificity of the method. Once these limits are clearly described, it would then make more sense to bring in a new method - capture - and describe its application in a situation where expected viral loads are below what metagenomics can deliver.

Specific comments:

None of the linked data or software seem to be available. Following the link to the

Premonition software, I could not find anywhere to download or explore the pipeline code. If the software is proprietary, this needs to be made clear; the paper suggests that the software should be available. The GitHub link is also broken (possibly the repo needs to be made public?).

Project PRJNA824010 contains only 3 assemblies (2x YFV, 1x Coxsackievirus B3), all three 'unverified'; and only 186 short read datasets. I was expecting 17 LASV genomes as per report, with additional genomes for other viruses.

Methods includes some results, eg.: L478-484 ("the two complete LASV genomes generated from the RT-qPCR-negative sample pool were identical. We checked for possible experimental contamination and determined that these samples were not processed in neighboring wells and were separated by intervening wells without LASV reads or partial genomes. Moreover, we assessed for possible epidemiological links" etc). -- This does not exclude contamination, either by droplets or as a sample mix-up. Were these genomes in a meaningful place on the phylogenetic tree to be consistent with sampling dates? Was there any other genome in this batch that was highly similar to these two?

Were unique dual indexes used? If not, the authors need to account for possible index misassignment, which tends to create sequence contamination even in the absence of real laboratory contamination. The distribution of LASV in PCR-ve samples (Figure 2B) implies a noisy method that requires a robust statistical framework to address - perhaps consider existing software designed for this, eg MetaMix. Two genomes that were PCR-negative but appeared identical may indeed be real, but this isn't convincing without a clear approach to controlling for contamination and index misassignment above

L150-152 "One LASV- sample was multiply co-infected, with anellovirus, LASV (i.e., this sample was one of the PCR false negatives that produced a full genome), and pegivirus C" – This type of observation is also consistent with a contaminated sample; at the very least, this cannot be ruled out from the presented results

L427-428 "Samples were required to have a greater percentage of reads assigned to a particular virus than the percentage of reads assigned to that virus across batch-specific water controls." -- Controls need to be of the same sample type as the samples being assessed (here: plasma), taken through the entire process from the earliest possible point, ideally pre-extraction. Comparisons with water controls are inadequate for metagenomics workflows; water controls do not have sufficient mass to produce a comparable library, and do not capture the whole process. As it's obviously impossible to include meaningful controls retrospectively (eg. verified negative plasma) the authors could address this partially for LASV using the PCR-negatives, with a statistical method that accounts for possible contamination of both PCR negatives and PCR positives, to determine a meaningful threshold for positivity. This can be done using an appropriate model that incorporates uncertainty in the labels, allowing for some negatives to be true positives (with a low prior) and some positives to be negatives (likewise). Failing that, at least a simple logistic regression would be helpful to assign a probability for each data point of being a positive, and then a reasonable threshold would need to be applied on what constitutes an

acceptable probability. Many other better methods are available - the stats as described relate only to groups and are not adequate for this. Please expand the methods and results appropriately.

Table 1 summarises samples in this study with PCR results, but it does not give enough information to summarise the actual experiments. As these are reasonably complex, please either expand this or provide an additional table with with the following columns:

- * Cohort
- * Numbers of patients and number of samples included in this study (not previously)
- * Number of samples tested by PCR
- * Numbers of PCR+ and PCR- samples positive and negative by metagenomics (without enrichment - ie not targeted capture), using an evidence-based threshold as outlined above to determine positivity
- * If targeted capture for MPX is included, then the number of PCR+ and PCR- samples positive and negative by targeted capture

L125-126 “Two samples failed to align any reads to the LASV reference genome, consistent with errors in metagenomic classification.” – This suggests a significant in-silico false discovery rate among the positives as well. What is the expected sensitivity and specificity of the Premonition pipeline?

L163 “Our work generated 17 new high quality (>90% of the genome 164 assembled) LASV genomes which we assembled and assessed” – I could not find these in the GenBank project listed

L165-167 “two 2018 false negatives” - please indicate these clearly on the tree in Fig3. Does the placement make sense relative to the date of collection, for example? Perhaps a time-resolved phylogeny may make this clearer, and give additional support to these being genuine missed positives

L184 HuBDV-2 only found in LASV+ samples, never in negatives – Given concerns about poor classifier performance (see above for two apparently non-LASV samples classified as LASV by the Premonition pipeline), can the authors exclude misalignment/misclassification of a divergent LASV virus? It is tantalising to speculate that the HuBDV-2 is real and is using LASV or another helper virus to replicate, but the far more prosaic possibility of bioinformatic failure needs to be robustly excluded.

L211 “We selected five MPXV+ samples – those with the highest sequencing library quantification values – for metagenomic sequencing, using pan-viral target enrichment probes due to the large genome size” — This is not metagenomic sequencing, this is targeted capture of MPXV. I don’t understand the relevance of this here, as the section title is “Metagenomics for ongoing surveillance and investigation of suspected outbreaks”: were the MPXV samples also sequenced metagenomically? It might be better to reframe this by pointout out that the MPXV viral loads were presumably below the expected limit of

detection by metagenomics, but were partially recoverably recoverable by a more sensitive method.

L328 MPXV is detectable in plasma in at least 49% of suspected cases, increasing the ease diagnostic testing relative to testing lesion samples. -- This is misleading as presented: MPXV was detected using PCR (not metagenomics as a cursory reading would suggest). Please add this clarification.

Can the authors expand the discussion of observed very high prevalence of Pegivirus C? This is a prevalence more usually associated with HIV+ populations, but detection of HIV in this dataset was quite low compared with Pegivirus.

Reviewer #2 (Remarks to the Author):

Key results

This study applied metagenomic characterization to 3 sources of clinical samples in Nigeria between 2017 to 2020 involving 612 febrile patients: 1) Lassa Fever/LASV PCR characterized samples, 2) 3 outbreak clusters of unknown etiology, and 3) 8 clinically-difficult cases of unknown etiology. The study includes several important findings, demonstrating the clinical utility of metagenomic sequencing at 91% concordance with LASV diagnostic testing; revealing several co-infections by health-threatening viruses previously unappreciated by way of circulation in Nigeria; attributing two of three outbreaks to YFV and monkeypox, respectively, and ruling out an infectious agent behind the third; and shedding light on potential infectious agents behind 2 of the 8 clinically challenging cases. The study produced 13 distinct virus genomes.

Validity

The data interpretation and conclusions are robust and valid. Metagenomic data is information rich, and the authors use rigorous protocols to ensure that genomic coverage is sufficient to warrant their conclusions, along with well-supported phylogenetic inference. They caution appropriately against drawing causal relationships between viruses discovered and clinical disease in the absence of being able to meet Kock's postulates.

Significance

The significance of these study results are unique and powerful, highlighting the potential clinical importance of co-infections as well as the presence of several viruses of significant health threat but previously unknown to Nigeria.

Data and methodology

The data are high quality, built on a prior study of over 500 LF patients as well as clinical and epidemiologic data collected in partnership with the NCDC. The metagenomics methods are sound and rigorous with the proper quality assessment and controls. The phylogenetic analyses are sound, although tree presentation in Fig. 4 should include support values, which were generated as described in the methods. Fig. 1 showing ii) study overview seems like a non-intuitive way to present the 3 epi clusters. The power of the methodology is the strong partnership between clinical, epidemiologic and diagnostic testing with

metagenomics approaches usually used mainly in research settings.

Analytical approach

Analyses including statistical tests, genome assembly, strain identification, and phylogenetic inference are all sound.

Suggested improvements

None, besides presentation suggestions noted above.

Clarity and context

Some suggestions to improve clarity/context

Consider using the new name for monkeypox – mpox: <https://www.who.int/news/item/28-11-2022-who-recommends-new-name-for-monkeypox-disease>

Line 216-217 – include whether MPVX was genetically identical in the same patient sampled by lesion versus metagenomically from plasma.

Line 341-342 – it would be interesting to reflect based on phylogenetic patterns across the different viruses whether circulation in Nigeria tends to be consistent with cryptic transmission versus importation of new strains.

Signed:

Shannon Bennett

California Academy of Sciences

Reviewer #3 (Remarks to the Author):

Oguzie et al used metagenomics to identify known and novel/divergent viruses from human infection samples in a Nigeria cohort. Using unbiased metagenomic sequencing, the authors investigated the viral etiologies for three different application scenarios, population-level LASV surveillance, cluster investigations of suspected pathogens, and diagnoses of unusual clinical cases. The paper also assessed and compared the detection performance between RT-qPCR testing and metagenomic sequencing, suggesting the usage of the latter method in unusual infection and co-infections. Phylogenetic analyses were conducted to resolve the evolutionary relationships of the detected pathogens. The Nigerian cohort is a great one, and the data generated from it is of very high scientific value. Overall, this is a nice work demonstrating the various utilities of metagenomics in disease investigation, and provides very valuable experience and troubleshooting solutions to potential problem that may come across in such metagenomic applications.

I have some suggestions below that hope to improve this manuscript:

1. Figure 2B: the authors performed Wilcoxon rank sum test to compare the number of LASV mapped reads between PCR-positive and negative samples. I feel that this part is to find evidence to explain the 11 PCR-negative but metagenomic-positive samples. For example, metagenomic reads of those PCR neg samples have far lower amount than the PCR pos. Therefore, it seems to me that the test shall compare PCR-positive and negative

samples using the subset of metagenomic-positive samples?

2. From line 121 to 123: the authors considered three possible reasons for the 11 PCR-negative but metagenomic-positive samples. I think there are other possible reasons worth to explore or at least discuss about.

-- 2a. Were any of these samples sequenced in Illumina platform in multiplexing manner? Since index hopping is a known problem in Illumina sequencing, if multiplex was applied then the index hopping would be a possible reason for the PCR-neg/Metag-pos samples. Depending on the techniques used in sequencing library construction, the hopping (and hence inter-flow of reads across samples) could be minimized in different extent. This issue can be investigated in good depth, e.g. whether those 11 samples have been multiplex sequenced in the same flow cell with other LASV positive samples; inconsistent dual barcodes have been checked and filtered, etc.

-- 2b. There is known read-carryover problem in certain Illumina sequencing platform, where reads from previous run appears in the subsequent run, especially if wash process between runs is not sufficient. It worths to investigate/rule out/mention this possible reason for seeing LASV reads in PCR-negative samples.

-- 2c. Line 125-126: It mentions that "Two samples failed to align any reads to the LASV reference genome, consistent with errors in metagenomic classification." The two samples refer to the two of the 11 PCR-neg/Metag-pos samples. It seems there is no details about how the metagenomic classification was done in the Method section; if so, would be useful to provide some details. At least this finding of erroneous metagenomic classification suggest that the classification method is not the same alignment method used for mapping the reads to the LASV genomes here. I am interested in how would this mis-classification happened, e.g. How many LASV reads were found in those two samples that eventually failed to align to LASV reference genome? If there are not singleton, it suggests the metagenomic classification would be better replaced with more accurate method.

3. The authors used metagenomic sequencing with five or more reads mapping to target pathogen group (LASV) as the criteria of metagenomic positive. However, in the line 426 to 434, the authors used both short reads and assembled contigs to define the presence of viruses. It is a bit confusing to me. Could the authors clarify this?

4. It would be useful to also present the co-infection pattern among all different viruses of concern (not just which virus co-infect with LASV patients) in those patient using a comprehensive heatmap or table (e.g. each row is a patient sample). This will provide useful epidemiology including other viruses.

5. The authors may perform deeper phylogenetic analyses such as temporal phylogeny and phylogeography to provide more insights about the virus evolution and transmission pattern at country and regional/global levels. This will enrich knowledge that can be gained from this great cohort, and new insights into the role of Nigeria in the emergence and transmission of virus of global concern.

6. The finding of dicistroviruses is really interesting. It would be helpful to readers with more detailed analysis on the dicistroviruses found in Nigeria. Phylogenetic analyses would be useful to at least show their evolutionary relationships with those found in other places.

7. The authors selected five MPXV+ samples with the best library quality for metagenomic sequencing. The authors may consider randomly select some among the 71 samples for sequencing since this could also verify the ability of MPXV detection in qPCR negative samples and detection of potential unknown clinical pathogens.

8. In the phylogenetic reconstruction section, I would suggest the authors to perform multiple sequence alignment separately for GPC and NP CDS regions before concatenating the aligned CDS, to avoid misalignment across two CDS regions.

9. Some very minor suggestions: a) it seems that some supplemental tables and figures were not labelled according to the order of their occurrences in the manuscript; b) it seems that the fonts in line 370-375 were not formatted consistently with the rest; c) Line 389, "QiAmp" shall be "QIAamp" I believe; d) Line 394 "ref." can be replaced by "Nikisins et al.".

Tommy Lam

We thank the Reviewers for their in-depth and helpful comments. Below is our point-by-point responses.
Thank you

Reviewer #1:

1. This work addresses a critical need for improved pan-pathogen surveillance in public health and clinical settings, and describes results using the most common method currently applied to this problem: pure/unbiased metagenomics. This is an expensive and laborious undertaking in any healthcare setting, and it is to the credit of the Nigerian authors that they were able to perform it in multiple contexts and on a large number of samples, albeit >200 of which have been previously reported. The work is potentially impactful and includes some novel findings - notably of a discovery of a new dicistrovirus and high prevalence of pegivirus C. However, there are a number of critical shortcomings in the methods, and some confusing reporting that should be addressed.

Thank you for emphasizing the importance of pathogen surveillance, our efforts to carry out extensive metagenomic sequencing in Africa, and the impact and novelty present in our findings. We hope that our revisions in response to your concerns clarify our approaches and bolster the rigor of our findings.

2. The findings are interesting but, having grouped together several cohorts and experiments, the authors have created a patchwork analysis that rests entirely on the strength of their methods. Unfortunately, the methods are either insufficiently thought out, or insufficiently described, and the experiments are missing essential process controls for contamination.

As is normal for metagenomics, the sequence data are noisy, with a lot of potential false positives among PCR-negative samples (eg Fig 2B), and presumably vice versa. This requires careful reporting. Finding true positives missed by PCR is an admirable goal - and I agree this is plausible particularly for diverse viruses - but it requires unequivocal demonstration that the metagenomics method delivers true positives and not a collection of spurious reads that are either contamination, index misassignment (aka index hopping), or misclassification. Figure 2B and much of the rest of the paper do not convince me that the authors have selected a meaningful lower limit of detection (5 reads), or that they have adequately minimised cross-detection (eg with unique dual indexes) or accounted for possible contamination (water controls are not enough).

Thank you for your feedback. We have extensively revised our presentation of our experimental and computational methods, including via a new results section detailing the steps that we use to call pathogens present in the paper.

In original Figures 2A and 2B, we defined “metagenomic positivity” as 5 or more reads mapped to the pathogen via Premonition. Given that we found that this threshold was not stringent enough, as you correctly note, for all other analyses in the paper, we require *both* 5 or more reads mapped to the pathogen via Premonition *and* pathogen genome assembly to yield contigs of at least 10% of the reference genome length. In light of this and other feedback, we have updated our analyses to use more stringent criteria throughout the entire paper. We also demonstrate the high specificity of this approach using Lassa virus PCR positivity as the “ground truth” (albeit an imperfect one, as noted in the text).

Because these experimental controls and computational processing steps are at the heart of a number of your questions, we provide the complete text here, and reference it in a number of answers below.

“The scale and complexity of metagenomic sequencing data, as well as the risk of contamination or pathogen mis-assignment, necessitate strict experimental and computational protocols to ensure that detected microbes are truly present. We developed procedures that greatly reduce the chance of calling false positives by i) using both negative and positive controls, ii) identifying intersample contamination, and iii) developing stringent bioinformatic procedures that prioritize specificity over sensitivity (Figure 1). Because our protocols evolved over the course of the study, we outline our recommendations and the proportion of the 593 total samples sequenced via metagenomics to which each procedure was applied (Supplementary Table 1).

Experimentally, we developed procedures to both mitigate the risk of and identify potential cases of contamination occurring in the laboratory. First, we extracted plasma samples in batches alongside non-template controls (i.e., water controls) for 574 (96.8%) samples. We designed batches to minimize the cases where samples known to be positive for a particular pathogen, such as Lassa virus (LASV), were extracted or sequenced with samples known to lack the pathogen. Before synthesizing cDNA or preparing sequencing libraries, we added a negative control (i.e., RNA isolated from K562 lymphoblast cells) and a positive control (i.e., RNA from viral seedstock spiked into RNA isolated from K562 lymphoblast cells, or RNA from a previously sequenced plasma sample known to contain a specific virus) for 585 (98.7%) and 509 (85.8%) samples, respectively. At this stage, we also added sample-specific RNA spike-ins using the External RNA Controls Consortium (ERCC) sequences for each of 508 (85.7%) samples, including all samples in batches of 12 or more, increasing the probability of detecting any downstream cross contamination.²⁰ We sequenced the majority of samples with combinatorial dual indexes (CDIs), although we used unique dual indexes (UDIs) for the one batch sequenced on the NovaSeq 6000 system (99 or 16.7% of samples) to minimize the risk of misclassification due to index hopping.

Computationally, we chose universal, strict filtering criteria to analyze the resulting data. We first discarded samples that displayed evidence of potential cross-contamination via the ERCC spike-ins (7 of 560 samples; Supplementary Figure 1A). We then ensured that the expected viral genomic material was identified in the positive controls via the metagenomic classification tool Microsoft Premonition²¹ (Supplementary Table 2). Next, to call a virus present in a sample, we required it to have (i) at least 5 reads assigned to it by Microsoft Premonition; (ii) a greater percent of reads assigned to it than assigned to the same species in any (a) extraction-batch-specific non-template control, (b) sequence-batch-specific positive control, excluding the spiked in viral genomic material, and (c) sequence-batch-specific negative control; and (iii) genome assembly of Microsoft Premonition hits with a threshold of at least 10% of the reference genome size (Supplementary Data, Supplementary Figure 2). Thus, we combined a highly sensitive, but less specific, probabilistic classification tool with a highly specific, but less sensitive contig assembly step to assign pathogens to samples.”

3. I also question the authors’ decision to conflate metagenomics with targeted capture; the latter is an excellent method and clearly was beneficial here for the MPXV samples, but it is distinct from unbiased metagenomics. Conflating the two looks like an attempt to elevate the success rate of unbiased metagenomics. Since this method only relates to the MPXV cohort, this adds further confusion to what is already quite a complex report. The authors should consider focusing specifically on unbiased metagenomics, as this is the bulk of the work, and expand the methods and results substantially to demonstrate sensitivity and specificity of the method. Once these limits are clearly described, it would then make more sense to bring in a new method - capture - and describe its application in a situation where expected viral loads are below what metagenomics can deliver.

Thank you for this feedback. We have restructured this section as suggested and now provide unbiased metagenomic data for the MPXV section, highlighting in the results and discussion that it was insufficient

to identify MPXV in qPCR-positive samples. We then demonstrate that the addition of a pan-viral hybrid capture step— which, though still biased, is not as limited in scope as single-pathogen hybrid capture— allowed us to generate MPXV contigs and to classify the MPXV clade as IIB. We include the following text:

“We selected five MPXV-positive plasma samples—those with the highest sequencing library quantification values—for unbiased sequencing as well as hybrid capture with pan-viral target enrichment probes (Methods). Unbiased metagenomics yielded 30 or fewer aligned read pairs for each sample, while hybrid capture yielded up to 20,000 aligned read pairs (Supplementary Figure 6). We produced contigs capable of determining that the 5 samples belonged to the IIB clade (i.e., the clade responsible for the 2022 multinational outbreak), consistent with other outbreak reports⁴⁷. We could not assemble complete genomes via either metagenomics or hybrid capture, likely due in part to the large genome size, reduced viral loads in the blood relative to lesions⁴⁹, and the Illumina MiSeq’s sequencing capacity.”

Specific comments:

4. None of the linked data or software seem to be available. Following the link to the Premonition software, I could not find anywhere to download or explore the pipeline code. If the software is proprietary, this needs to be made clear; the paper suggests that the software should be available. The GitHub link is also broken (possibly the repo needs to be made public?).

The Microsoft Premonition team will make the software available ahead of the planned public release to individuals who reach out with regards to this work. We have added the following line to the text:

“Information about the Microsoft Premonition metagenomics pipeline is available at <https://microsoft.com/premonition>. Individuals can access the pipeline ahead of its public release by clicking the “Contact us for availability” button and mentioning this work, or by emailing Simon Frost at Frost.Simon@microsoft.com.”

We have also confirmed that the GitHub link works on multiple computers, including those of individuals without GitHub accounts. Please let us know if this problem persists.

5. Project PRJNA824010 contains only 3 assemblies (2x YFV, 1x Coxsackievirus B3), all three ‘unverified’; and only 186 short read datasets. I was expecting 17 LASV genomes as per report, with additional genomes for other viruses.

We completed the process of submitting short read datasets and genomes to NCBI GenBank as the work was under review and revisions. We are waiting for NCBI to provide genome accession numbers, though the BioSample identifiers (and the associated SRA data) are listed in the Supplementary Data File. The datasets and genomes fall under 2 distinct BioProjects, as some samples were registered with the publication of Siddle et al. 2018. We have added the following line to the text:

“We deposited raw reads to the Sequence Read Archive (SRA) and assembled genomes that were at least 80% complete to GenBank under BioProjects PRJNA824010 and PRJNA436552.”

6. Methods includes some results, eg.: L478-484 (“the two complete LASV genomes generated from the RT-qPCR-negative sample pool were identical. We checked for possible experimental contamination and determined that these samples were not processed in neighboring wells and were

separated by intervening wells without LASV reads or partial genomes. Moreover, we assessed for possible epidemiological links” etc). -- This does not exclude contamination, either by droplets or as a sample mix-up. Were these genomes in a meaningful place on the phylogenetic tree to be consistent with sampling dates? Was there any other genome in this batch that was highly similar to these two?

We agree that this result warrants greater detail about how we rule out experimental contamination. As we described in our response to Point 2, we added additional text to describe our experimental and bioinformatic processes to ensure high quality results. Specifically to LASV-positive results in the RT-qPCR-negative samples, we have also updated the text to include the respective information in the results section and in a supplementary note:

*“The imperfect specificity was attributable to 4 samples that were RT-qPCR-negative but positive via sequencing. Two of these samples yielded complete, identical LASV genomes (98% and 99% complete) while the other two samples yielded partial genomes. We extensively queried these samples and re-tested them via RT-qPCR (**Supplementary Note, Supplementary Figure 3**), ultimately concluding that they were diagnostic false negatives, a known challenge in LASV molecular detection²⁶.”*

“We detected Lassa virus (LASV) via metagenomics in 4 samples that were negative for LASV via clinical RT-qPCR testing. To ensure that these samples were true false negatives, we thoroughly investigated their provenance.

- *We re-tested the samples via RT-qPCR following metagenomic sequencing, and confirmed that they were RT-qPCR-negative (**Supplementary Figure 3**).*
- *We confirmed that the ERCC RNA spike-ins were highly pure for these samples (>1.4 million reads assigned to ERCCs, of which >99.97% were assigned to the proper ERCC). This suggests that interwell contamination cannot explain these findings.*
- *We sequenced the samples using unique dual indexes, minimizing the likelihood of index hopping. They were sequenced on a different sequencing machine than the RT-qPCR-positive samples in our study, 4 months after any RT-qPCR-positive samples were processed in the laboratory.*
- *We compared the identical, complete LASV genomes that we produced from 2 of these samples to all LASV genomes present in NCBI GenBank and in our study. They were genetically distinct from all other available genomes.*
- *We analyzed the 4 genomes for mutations in the regions mapping to the Nikisins primers, which target the L gene²³. The 2 partial genomes lacked coverage in the primer-binding regions. The 2 complete genomes possessed a mismatch at the second position of the forward primer (CAACCATYTTTRTGATRTGCCA).*
- *We queried for epidemiological links between the 2 individuals with identical LASV genomes. Their samples were collected 2 days apart, though the individuals reside in different states and age brackets. However, we could not comprehensively rule out the possibility of human-to-human transmission.”*

We assessed whether these genomes were found in a reasonable location on the phylogenetic tree, given their sampling locations. The genomes were of lineage II, which is expected because the samples were collected from individuals living in Imo and Edo, in southern Nigeria. We have added the following line to the text:

“Most of our genomes, including those from the PCR-negative samples, were of lineage II, and clustered according to their sampling site (Irrua in the southwestern cluster and Ebonyi in the southeastern cluster).”

7. Were unique dual indexes used? If not, the authors need to account for possible index misassignment, which tends to create sequence contamination even in the absence of real laboratory contamination. The distribution of LASV in PCR-ve samples (Figure 2B) implies a noisy method that requires a robust statistical framework to address - perhaps consider existing software designed for this, eg MetaMix. Two genomes that were PCR-negative but appeared identical may indeed be real, but this isn't convincing without a clear approach to controlling for contamination and index misassignment above

We used unique dual indexes for the LASV RT-qPCR-negative samples. The LASV-positive samples were not sequenced with UDIs, but were sequenced on instruments in which index misassignment is rare (HiSeq2500, MiSeq). We now note this in the results section and have added a new table (**Supplementary Table 1**) to specify per-batch indexes and which batches were sequenced with ERCC RNA spike-ins.

8. L150-152 “One LASV- sample was multiply co-infected, with anellovirus, LASV (i.e., this sample was one of the PCR false negatives that produced a full genome), and pegivirus C” – This type of observation is also consistent with a contaminated sample; at the very least, this cannot be ruled out from the presented results

Thank you for this feedback. We agree that such cases warrant detailed consideration and believe our revised presentation of the stringent protocols we followed—including negative and positive controls, spike-ins, and a 10% genome assembly length threshold—indicate that contamination is unlikely. For this specific case, several findings point to this result being accurate:

- Spike-in data for this sample was extremely clean, with >99.99% of over 15 million reads mapping to a single spike-in sequence.
- Only one other sample across the nearly 600 samples sequenced in this study contained anellovirus. The multiply-coinfected sample was sequenced in the United States in October 2018, while the other sample was sequenced in Nigeria in January 2020 (**Supplementary Figure 2**).
- The pegivirus C genome produced from this sample was not identical to any other sequences in this study, and was more closely related to 2 genomes from Cameroon than from >90% of the other pegivirus C genomes produced in this study (**Supplementary Figure 5**).

9. L427-428 “Samples were required to have a greater percentage of reads assigned to a particular virus than the percentage of reads assigned to that virus across batch-specific water controls.” -- Controls need to be of the same sample type as the samples being assessed (here: plasma), taken through the entire process from the earliest possible point, ideally pre-extraction. Comparisons with water controls are inadequate for metagenomics workflows; water controls do not have sufficient mass to produce a comparable library, and do not capture the whole process. As it's obviously impossible to include meaningful controls retrospectively (eg. verified negative plasma) the authors could address this partially for LASV using the PCR-negatives, with a statistical method that accounts for possible contamination of both PCR negatives and PCR positives, to determine a meaningful threshold for positivity. This can be done using an appropriate model that incorporates uncertainty in the labels, allowing for some negatives to be true positives (with a low prior) and some positives to be negatives (likewise). Failing that, at least a simple logistic regression would be helpful to assign a probability for each data point of being a positive, and then a reasonable threshold would need to be applied on what constitutes an acceptable probability. Many other better methods are available - the stats as described relate only to groups and are not adequate for this. Please expand the methods and results appropriately.

We thank the reviewer for these excellent points. We used water controls during extraction, followed by K562 (lymphoblast cell line) RNA as a negative control capable of producing a comparable library. As described in our response to Point 2, we describe our experimental controls and bioinformatic processing steps further in our new results section.

10. Table 1 summarises samples in this study with PCR results, but it does not give enough information to summarise the actual experiments. As these are reasonably complex, please either expand this or provide an additional table with the following columns:

* Cohort

* Numbers of patients and number of samples included in this study (not previously)

* Number of samples tested by PCR

* Numbers of PCR+ and PCR- samples positive and negative by metagenomics (without enrichment - ie not targeted capture), using an evidence-based threshold as outlined above to determine positivity

* If targeted capture for MPX is included, then the number of PCR+ and PCR- samples positive and negative by targeted capture

The desired information for all cohorts can be found in the **Supplementary Data File**. We have restricted **Table 1** to the description of only the cohorts suspected of Lassa Fever (LF). We believe that the updated **Figure 1** clarifies that, in addition to samples from population-level surveillance of LF-like illness, there are additionally three outbreak-associated cohorts, as well as eight critically ill individuals who are further described in **Supplementary Table 7**.

11. L125-126 “Two samples failed to align any reads to the LASV reference genome, consistent with errors in metagenomic classification.” – This suggests a significant in-silico false discovery rate among the positives as well. What is the expected sensitivity and specificity of the Premonition pipeline?

We assessed the sensitivity and specificity of the Premonition pipeline, using LASV RT-qPCR status as the ground truth (albeit an imperfect one). When solely using 5 reads mapped to LASV via Premonition as the threshold to call a virus present, the pipeline had a sensitivity of 91.7% and a specificity of 91.6%.

Because we desired higher specificity and prioritized it over sensitivity, we adjusted our criteria to include both 5 reads mapped to LASV via Premonition and the assembly of a LASV genome of minimally 10% of the reference genome length. These criteria resulted in a sensitivity of 35.4% and a specificity of minimally 95.8%. Note that we believe this specificity to be higher, between 97.9% and 100%, given i) the assembly of complete genomes from 2 RT-qPCR false negatives and partial genomes from 2 RT-qPCR false negatives; and ii) a known LASV diagnostic false negative rate of 5-10% (Nikisins et al. 2015; PLoS Negl Trop Dis). We have added the following line to the text:

*“We assessed the sensitivity and specificity of our metagenomic pipeline relative to clinical RT-qPCR by using data from the cohort of individuals suspected of Lassa Fever (LF). A positive Lassa virus (LASV) clinical test was defined as the amplification of either the GPC gene or the L gene via the commercially available Altona assay^{22,23}. Prior clinical RT-qPCR status is an imperfect ground truth, as i) genome degradation can occur between clinical testing and subsequent sequencing and ii) RT-qPCR can yield false negative results for samples containing highly diverse viruses, such as LASV. Moreover, we expect PCR to be more sensitive than metagenomics due to target amplification^{24,25}. Nevertheless, we found that the Premonition-based thresholds yielded a sensitivity of 91.7% and a specificity of 91.6%; the additional requirement of contig assembly reduced sensitivity to 35.4% but increased specificity to minimally 95.8% (**Supplementary Figure 1B**). The imperfect specificity was attributable to 4 samples that were RT-qPCR-negative but positive via sequencing. Two of these samples yielded complete, identical LASV*

genomes (98% and 99% complete) while the other two samples yielded partial genomes. We extensively queried these samples and re-tested them via RT-qPCR (**Supplementary Note, Supplementary Figure 3**), ultimately concluding that they were diagnostic false negatives, a known challenge in LASV molecular detection²⁶. In summary, our metagenomic protocols demonstrated high specificity for identifying pathogens in a given sample.”

12. L163 “Our work generated 17 new high quality (>90% of the genome 164 assembled) LASV genomes which we assembled and assessed” – I could not find these in the GenBank project listed

Thank you for this reminder. We completed the process of submitting short read datasets and genomes to NCBI GenBank as the work was under revisions. These specific genomes can be found under BioProject PRJNA824010.

13. L165-167 “two 2018 false negatives” - please indicate these clearly on the tree in Fig3. Does the placement make sense relative to the date of collection, for example? Perhaps a time-resolved phylogeny may make this clearer, and give additional support to these being genuine missed positives

We have indicated the false negatives clearly on the tree. Their placement makes sense relative to the date of collection as well as the location of collection. They belong to clade II, which is consistent with their geographic origins in Imo and Edo. Perhaps the most compelling evidence that they are genuine missed positives is the fact that they are non-identical to any other publicly available genomes.

14. L184 HuBDV-2 only found in LASV+ samples, never in negatives – Given concerns about poor classifier performance (see above for two apparently non-LASV samples classified as LASV by the Premonition pipeline), can the authors exclude misalignment/misclassification of a divergent LASV virus? It is tantalising to speculate that the HuBDV-2 is real and is using LASV or another helper virus to replicate, but the far more prosaic possibility of bioinformatic failure needs to be robustly excluded.

We conducted reference-based genome assembly using full-length dicistrovirus genomes assembled from the plasma of febrile Tanzanian children (DOI: [10.1080/22221751.2019.1603791](https://doi.org/10.1080/22221751.2019.1603791)), and assembled genomes that had >96% nucleotide identity to these genomes from Tanzania. The resulting genomes, which we name HuBDV-2, are approximately 9 kb. We used MAFFT v7.490 to align one of our HuBDV-2 genomes to the 3 kb (NC_004296.1) and 7 kb (NC_004297.1) LASV segments, finding that Hu-BDV shares 33.0% identity to the LASV L segment and 19.3% identity to the LASV S segment. While misclassification is an important consideration, it is highly unlikely that these HuBDV-2 genomes resulted from the misclassification of a divergent LASV.

We also want to note that our detection of HuBDV-2 in LASV-positive samples, but not LASV-negative samples, may be explained entirely by sample size. We suspect that HuBDV-2 infection is rare– we find it in 2/458 LASV-positive samples and 0/95 LASV-negative samples, or 0.36% of total samples suspected of LF. We sequenced over four times as many LASV-positive samples as LASV-negative samples. More data would be needed to see if HuBDV-2 is significantly enriched in LASV-positive samples.

Given that the dicistroviruses represent an important finding from our work, we have conducted additional phylogenetic analyses of the HuBDV and HuBDV-2 genomes. We have copied **Figure 4** below:

Figure 4. *Dicistrovirus RdRp (RNA-dependent RNA polymerase) genetic diversity. Maximum likelihood phylogenetic tree with 3 new sequences (green) alongside 21 published sequences. Bootstrap values and the clade that we name human blood-associated dicistrovirus 2 (HuBDV-2) are shown.*

15. L211 “We selected five MPXV+ samples – those with the highest sequencing library quantification values – for metagenomic sequencing, using pan-viral target enrichment probes due to the large genome size” — This is not metagenomic sequencing, this is targeted capture of MPXV. I don’t understand the relevance of this here, as the section title is “Metagenomics for ongoing surveillance and investigation of suspected outbreaks”: were the MPXV samples also sequenced metagenomically? It might be better to reframe this by point out that the MPXV viral loads were presumably below the expected limit of detection by metagenomics, but were partially recoverable by a more sensitive method.

We have now rewritten this section to make it clear that the MPXV study involved qPCR, hybrid capture, and metagenomic sequencing, and describe our findings using each method, as quoted in the response to Point 3.

16. L328 MPXV is detectable in plasma in at least 49% of suspected cases, increasing the ease diagnostic testing relative to testing lesion samples. -- This is misleading as presented: MPXV was detected using PCR (not metagenomics as a cursory reading would suggest). Please add this clarification.

We agree that the previous MPXV section lacked clarity in describing what we identified through qPCR, hybrid capture, and metagenomic sequencing. We added text in the results (quoted in response to Point 3) and the discussion sections to add clarity:

“Our MPXV investigation demonstrated the benefit of targeted approaches (e.g., qPCR and hybrid capture) when a pathogen is suspected, while also demonstrating that MPXV can be detected and subtyped from plasma samples.”

17. Can the authors expand the discussion of observed very high prevalence of Pegivirus C? This is a prevalence more usually associated with HIV+ populations, but detection of HIV in this dataset was quite low compared with Pegivirus.

Thank you for this feedback. The prevalence of pegivirus C in our study is consistent with what is seen in healthy individuals in West Africa. We have added the following line to the discussion before discussing the over-representation of pegivirus C in Lassa-positive cases that survived.

“Over 5% of the LASV-positive cohort was co-infected with pegivirus C, which is consistent with its estimated prevalence of 7-12% in healthy West African blood donors^{68,69}.”

Reviewer #2:

Key results

This study applied metagenomic characterization to 3 sources of clinical samples in Nigeria between 2017 to 2020 involving 612 febrile patients: 1) Lassa Fever/LASV PCR characterized samples, 2) 3 outbreak clusters of unknown etiology, and 3) 8 clinically-difficult cases of unknown etiology. The study includes several important findings, demonstrating the clinical utility of metagenomic sequencing at 91% concordance with LASV diagnostic testing; revealing several co-infections by health-threatening viruses previously unappreciated by way of circulation in Nigeria; attributing two of three outbreaks to YFV and monkeypox, respectively, and ruling out an infectious agent behind the third; and shedding light on potential infectious agents behind 2 of the 8 clinically challenging cases. The study produced 13 distinct virus genomes.

Validity

The data interpretation and conclusions are robust and valid. Metagenomic data is information rich, and the authors use rigorous protocols to ensure that genomic coverage is sufficient to warrant their conclusions, along with well-supported phylogenetic inference. They caution appropriately against drawing causal relationships between viruses discovered and clinical disease in the absence of being able to meet Kock’s postulates.

Significance

The significance of these study results are unique and powerful, highlighting the potential clinical importance of co-infections as well as the presence of several viruses of significant health threat but previously unknown to Nigeria.

Data and methodology

The data are high quality, built on a prior study of over 500 LF patients as well as clinical and epidemiologic data collected in partnership with the NCDC. The metagenomics methods are sound and rigorous with the proper quality assessment and controls. The phylogenetic analyses are sound,

although tree presentation in Fig. 4 should include support values, which were generated as described in the methods.

Thank you for your positive assessment of our methods, results, and conclusions. We have added bootstrap support values to the trees in Figure 4 (now **Figure 5**). We have copied it here for ease of review.

A**B**
Figure 5. A. *Coxsackievirus B3 (CV-B3)* genetic diversity. Maximum likelihood phylogenetic tree with one new sequence (green) alongside 63 full-length, published sequences. Bootstrap values are shown. **B.** *Hepatitis A* genetic diversity. Maximum likelihood phylogenetic tree with 2 new sequences (green) alongside 105 full-length, published sequences. Bootstrap values are shown. X-axes, number of mutations.

Fig. 1 showing ii) study overview seems like a non-intuitive way to present the 3 epi clusters. The power of the methodology is the strong partnership between clinical, epidemiologic and diagnostic testing with metagenomics approaches usually used mainly in research settings.

Thank you for this excellent point. We have altered **Figure 1**, copied below, to more clearly delineate these three clusters.

Figure 1: Overview of the study design. We conducted RT-qPCR and metagenomic sequencing on 670 plasma samples received from: (i) individuals suspected to have Lassa Fever (LF; caused by Lassa virus, LASV), collected from teaching hospitals with clinical expertise in viral hemorrhagic fevers; (ii) suspected infectious disease outbreaks, collected by the Nigerian Centre for Disease Control (NCDC) and other regional clinics; and (iii) individuals with unusual or nonspecific clinical manifestations from regional clinics. We used a metagenomic pipeline inspired by Matranga et al.¹ with additional negative (i.e., water and K562 cells) and positive controls (i.e., K562 cells spiked with known viral genetic material), as well as External RNA Controls Consortium (ERCC) RNA spike-ins (Thermo Fisher Scientific). We use metagenomics to identify putative causes of Lassa-like illness, to assess the role of co-infection in LASV outcomes, to determine the relationships between clinically similar acute illnesses, and to diagnose individuals with nonspecific presentations. Created with BioRender.com.

Analytical approach

Analyses including statistical tests, genome assembly, strain identification, and phylogenetic inference are all sound.

Suggested improvements

None, besides presentation suggestions noted above.

Clarity and context

Some suggestions to improve clarity/context

Consider using the new name for monkeypox – mpox:
<https://www.who.int/news/item/28-11-2022-who-recommends-new-name-for-monkeypox-disease>

Thank you. We have changed the syndrome to mpox throughout. We also use the WHO's revised names for viral lineages (clades I and II) and refer to the virus as monkeypox virus or MPXV in the text and figures.

Line 216-217 – include whether MPVX was genetically identical in the same patient sampled by lesion versus metagenomically from plasma.

In this study, we only had access to plasma samples containing MPXV. We have clarified this point in the following line:

“We selected five MPXV-positive plasma samples—those with the highest sequencing library quantification values—for unbiased sequencing as well as hybrid capture with pan-viral target enrichment probes (Methods).”

Line 341-342 – it would be interesting to reflect based on phylogenetic patterns across the different viruses whether circulation in Nigeria tends to be consistent with cryptic transmission versus importation of new strains.

Signed:

Shannon Bennett

California Academy of Sciences

Thank you for this feedback. We have added the following paragraph to the discussion section:

“Our phylogenetic reconstructions also produced actionable public health insights. Our finding that 2020 YFV cases were descendants of 2018 Nigerian cases indicated the presence of cryptic transmission and prompted the NCDC and the National Primary Health Care Development Authority (NHPCDA) to accelerate their vaccination efforts. On the other hand, our pegivirus C genomes were interspersed with those from Cameroon, Sierra Leone, and Uganda, emphasizing that transmission patterns in Nigeria are a result of both importation and internal circulation. Finally, we identified undersampled viral diversity, both by sequencing LASV samples from Kebbi, which form a clade within lineage III, and by generating the first complete coxsackievirus B3 genome on the African continent.”

Reviewer #3:

Oguzie et al used metagenomics to identify known and novel/divergent viruses from human infection samples in a Nigeria cohort. Using unbiased metagenomic sequencing, the authors investigated the viral etiologies for three different application scenarios, population-level LASV surveillance, cluster investigations of suspected pathogens, and diagnoses of unusual clinical cases. The paper also assessed and compared the detection performance between RT-qPCR testing and metagenomic sequencing,

suggesting the usage of the latter method in unusual infection and co-infections. Phylogenetic analyses were conducted to resolve the evolutionary relationships of the detected pathogens. The Nigerian cohort is a great one, and the data generated from it is of very high scientific value. Overall, this is a nice work demonstrating the various utilities of metagenomics in disease investigation, and provides very valuable experience and troubleshooting solutions to potential problem that may come across in such metagenomic applications.

Thank you for your positive view of our work, and for emphasizing the important takeaways of our study, both for public health in Nigeria and for scientists interested in applying metagenomics to different cohorts.

I have some suggestions below that hope to improve this manuscript:

1. Figure 2B: the authors performed Wilcoxon rank sum test to compare the number of LASV mapped reads between PCR-positive and negative samples. I feel that this part is to find evidence to explain the 11 PCR-negative but metagenomic-positive samples. For example, metagenomic reads of those PCR neg samples have far lower amount than the PCR pos. Therefore, it seems to me that the test shall compare PCR-positive and negative samples using the subset of metagenomic-positive samples?

Thank you for your feedback. We have re-written the entire first section of the manuscript to more clearly describe our experimental and computational methods. We also include this text in Reviewer 1, Point 2. We have removed the analysis previously reported in Figure 2B, as we now require both metagenomic classification of 5 or more reads and contigs covering 10% of the reference genome length as our criteria to call a pathogen present throughout the manuscript. These criteria prioritized specificity over sensitivity and resulted in 4 PCR-negative, metagenomic-positive samples (extensively investigated in **Supplementary Note**).

2. From line 121 to 123: the authors considered three possible reasons for the 11 PCR-negative but metagenomic-positive samples. I think there are other possible reasons worth to explore or at least discuss about.

-- 2a. Were any of these samples sequenced in Illumina platform in multiplexing manner? Since index hopping is a known problem in Illumina sequencing, if multiplex was applied then the index hopping would be a possible reason for the PCR-neg/Metag-pos samples. Depending on the techniques used in sequencing library construction, the hopping (and hence inter-flow of reads across samples) could be minimized in different extent. This issue can be investigated in good depth, e.g. whether those 11 samples have been multiplex sequenced in the same flow cell with other LASV positive samples; inconsistent dual barcodes have been checked and filtered, etc.

We have extensively investigated these cases (**Supplementary Note**), in addition to our more rigorous filtering that reduces the number from 11 to 4. While the RT-qPCR-negative samples were pooled and sequenced together on the same sequencing run, no LASV-positive samples were prepared or sequenced with these samples. We used dual unique indexes with stringent demultiplexing parameters, so we do not believe that index hopping is contributing to these findings. We also used sample-specific spike-in controls to monitor for the possibility of cross-contamination during sample preparation and did not observe evidence of any contamination in these samples. We now describe these steps to minimize erroneous assignments in a new section at the beginning of the results.

-- 2b. There is known read-carryover problem in certain Illumina sequencing platform, where reads from previous run appears in the subsequent run, especially if wash process between runs is not sufficient. It worths to investigate/rule out/mention this possible reason for seeing LASV reads in PCR-negative samples.

Thank you for noting this possibility. We have added **Supplementary Table 1**, which describes the sequencing processes and controls used for different batches of samples. Notably, the PCR-negative samples were sequenced on a different machine than the LASV-positive samples. Moreover, the PCR-negative samples were sequenced at the Broad Institute at least 4 months after any LASV-positive samples were sequenced at the Broad Institute. Therefore, we consider this possibility highly unlikely.

-- 2c. Line 125-126: It mentions that “Two samples failed to align any reads to the LASV reference genome, consistent with errors in metagenomic classification.” The two samples refer to the two of the 11 PCR-neg/Metag-pos samples. It seems there is no details about how the metagenomic classification was done in the Method section; if so, would be useful to provide some details. At least this finding of erroneous metagenomic classification suggest that the classification method is not the same alignment method used for mapping the reads to the LASV genomes here. I am interested in how would this mis-classification happened, e.g. How many LASV reads were found in those two samples that eventually failed to align to LASV reference genome? If there are not singleton, it suggests the metagenomic classification would be better replaced with more accurate method.

Thank you for this feedback. In addressing the comments of all reviewers, we simplified the presentation of the data. These 2 samples had 5 or more reads mapped to LASV via Premonition, but did not meet the 10% genome assembly threshold. We believe that this highlights the importance of using orthogonal tools to improve specificity, as described in Reviewer 1, Point 2.

Importantly, these 2 samples were among the 7 total samples removed from our sample set because their ERCC spike-in purity scores were below our thresholds, demonstrating evidence of potential cross-contamination.

3. The authors used metagenomic sequencing with five or more reads mapping to target pathogen group (LASV) as the criteria of metagenomic positive. However, in the line 426 to 434, the authors used both short reads and assembled contigs to define the presence of viruses. It is a bit confusing to me. Could the authors clarify this?

We apologize for the confusion, which we hope is clarified in our new results section and in our answer to Reviewer 1, Point 2. We are using both short read metagenomic classification and assembly of contigs to define the presence of viruses in the revised manuscript.

4. It would be useful to also present the co-infection pattern among all different viruses of concern (not just which virus co-infect with LASV patients) in those patient using a comprehensive heatmap or table (e.g. each row is a patient sample). This will provide useful epidemiology including other viruses.

Thank you for this suggestion. We have included these data as a heat map in **Supplementary Figure 2**.

5. The authors may perform deeper phylogenetic analyses such as temporal phylogeny and phylogeography to provide more insights about the virus evolution and transmission pattern at country and regional/global levels. This will enrich knowledge that can be gained from this great cohort, and new insights into the role of Nigeria in the emergence and transmission of virus of global concern.

Thank you for this suggestion. We have added phylogenies for the dicistroviruses (**Figure 4**) and Pegivirus C (**Supplementary Figure 5**), and have updated our phylogenetic analyses for coxsackievirus B3 (**Figure 5**), hepatovirus A (**Figure 5**), LASV (**Figure 3**), and YFV (**Supplementary Figure 7**). We also added the following paragraph to the discussion section:

“Our phylogenetic reconstructions also produced actionable public health insights. Our finding that 2020 YFV cases were descendants of 2018 Nigerian cases indicated the presence of cryptic transmission and prompted the NCDC and the National Primary Health Care Development Authority (NHPCDA) to accelerate their vaccination efforts. On the other hand, our pegivirus C genomes were interspersed with those from Cameroon, Sierra Leone, and Uganda, emphasizing that transmission patterns in Nigeria are a result of both importation and internal circulation. Finally, we identified undersampled viral diversity, both by sequencing LASV samples from Kebbi, which form a clade within lineage III, and by generating the first complete coxsackievirus B3 genome on the African continent.”

6. The finding of dicistroviruses is really interesting. It would be helpful to readers with more detailed analysis on the dicistroviruses found in Nigeria. Phylogenetic analyses would be useful to at least show their evolutionary relationships with those found in other places.

Thank you for this suggestion. We have included a phylogenetic analysis of the RNA-dependent RNA polymerase (RdRp) gene of the dicistrovirus family (**Figure 4**, in Reviewer 1, Point 14), including the Peruvian and Tanzanian human-derived sequences, sequences derived from arthropod hosts, and our three full-length sequences. We have denoted the clade that we call human blood-associated dicistrovirus 2 (HuBDV-2).

7. The authors selected five MPXV+ samples with the best library quality for metagenomic sequencing. The authors may consider randomly select some among the 71 samples for sequencing since this could also verify the ability of MPXV detection in qPCR negative samples and detection of potential unknown clinical pathogens.

We achieved relatively low coverage of the MPXV genome via both unbiased metagenomics and pan-viral hybrid capture, even when selecting the 5 samples with the best library quantifications (**Supplementary Figure 6**, in Reviewer 1, Point 3) out of the 35 that tested positive by qPCR. This suggests that qPCR diagnostics are far more sensitive than sequencing for detecting MPXV in plasma samples. We thus do not expect to find MPXV in the diagnostic negative samples, even if they are true positives.

8. In the phylogenetic reconstruction section, I would suggest the authors to perform multiple sequence alignment separately for GPC and NP CDS regions before concatenating the aligned CDS, to avoid misalignment across two CDS regions.

Thank you for this feedback. To confirm this could not have impacted our results, we visually inspected the alignment to ensure that the stop codon for the GPC gene and the start codon for the NP gene were

properly aligned. We then performed the alignment as suggested, prior to concatenating the aligned CDS regions, and ultimately obtained the same alignment.

9. Some very minor suggestions: a) it seems that some supplemental tables and figures were not labelled according to the order of their occurrences in the manuscript; b) it seems that the fonts in line 370-375 were not formatted consistently with the rest; c) Line 389, “QiAmp” shall be “QIAamp” I believe; d) Line 394 “ref.” can be replaced by “Nikisins et al.”.

Tommy Lam

Thank you for your careful reading of our text.

- a) We have updated the supplementary table and figure labels to follow the order of their occurrences in the manuscript.
- b) We have ensured that the font is consistent throughout the text.
- c) We have changed “QiAmp” to “QIAamp.”
- d) We have replaced “ref. 24” with “Nikisins et al.”

REVIEWERS' COMMENTS

Reviewer #1 (Remarks to the Author):

The authors have substantially revised and improved the manuscript and have addressed all major comments I raised, particularly regarding the methods. They have also uploaded further genomes to the SRA and the project PRJNA824010 now has a reasonable number of genomes (although this still seems to be less than reported, but I trust the authors will cross-check and complete this as needed prior to publication).

The authors have added an entirely new section on a putative causal link between pegivirus C positivity and reduced Ct in lassa virus infections. An obvious alternative is that pegivirus C co-infections is only detectable when the load of lassa virus is low, due to competition for reads. Can the authors comment on this possibility and any implications?

I can confirm that the github repository <https://github.com/bpetros95/lassa-metagenomics> is available; the issue I encountered previously was related to the way the link is encoded in the PDF (the page number seemed to have been included in the underlying URL). This is presumably going to be fixed in the final typeset version.

Reviewer #2 (Remarks to the Author):

The authors responded to all of my comments appropriately. The manuscript is of excellent quality and the study contributes critical understanding to emerging and re-emerging infectious diseases in Nigeria.

I have only two final edits:

Figure 4 – scale bar should include units

Figure 5 – x-axis should be labeled

Signed:

Shannon Bennett

California Academy of Sciences

San Francisco, CA

Reviewer #3 (Remarks to the Author):

The authors have fully addressed my previous review comments. I only have very minor suggestion for further improvement on their revision.

Fig S7. It would be more informative to use phylogram (branch length in the unit of genetic distance) instead of cladogram. Also, clearer if the length of sequence, unit of the tree branch, which genes are used to build the tree is indicated in the figure legend. Same comments on Figs 4,5 and Fig S5.

Tommy Lam

Reviewer #1

The authors have substantially revised and improved the manuscript and have addressed all major comments I raised, particularly regarding the methods. They have also uploaded further genomes to the SRA and the project PRJNA824010 now has a reasonable number of genomes (although this still seems to be less than reported, but I trust the authors will cross-check and complete this as needed prior to publication).

Thank you for your rigorous review, which greatly improved the presentation of our work. We have uploaded all short reads to the SRA. We uploaded all near-complete genomes (minimally 80% complete with sufficient coding sequence coverage for proper annotation) to GenBank under bioprojects PRJNA824010 and PRJNA436552. Accession numbers for both reads, and genomes are provided in the Supplementary Data File (under the “samples” and “viruses_present” tabs, respectively). In additional

The authors have added an entirely new section on a putative causal link between pegivirus C positivity and reduced Ct in lassa virus infections. An obvious alternative is that pegivirus C co-infections is only detectable when the load of lassa virus is low, due to competition for reads. Can the authors comment on this possibility and any implications?

This is a great question and one that we considered prior to conducting the analysis. If pegivirus C and Lassa virus (LASV) were competing for reads, we would expect to see enrichment of pegivirus C in the cohort without LASV. Instead, we report pegivirus C in 25/458 (5.46%) of LASV-positive individuals and 5/91 (5.49%) of LASV-negative individuals (Figure 2A). Moreover, we found no relationship between the number of reads assigned to LASV and the number of reads assigned to pegivirus C across all samples (panel added to **Supplementary Figure 4**; $r = -0.02$, $p = 0.63$).

We have added the following line to the text: “Importantly, we confirmed that there was no relationship between pegivirus C and LASV detection, i.e., due to competition for sequencing reads (**Figure 2A**; **Supplementary Figure 4J**).”

I can confirm that the github repository <https://github.com/bpetros95/lassa-metagenomics> is available; the issue I encountered previously was related to the way the link is encoded in the PDF (the page number seemed to have been included in the underlying URL). This is presumably going to be fixed in the final typeset version.

Thank you for checking that the code is publicly available. The page numbers will not appear in the final typeset version.

Reviewer #2

The authors responded to all of my comments appropriately. The manuscript is of excellent quality and the study contributes critical understanding to emerging and re-emerging infectious diseases in Nigeria.

I have only two final edits:

Figure 4 – scale bar should include units

Figure 5 – x-axis should be labeled

Signed:

Shannon Bennett

California Academy of Sciences

San Francisco, CA

Thank you for your feedback. We have added units to the scale bar in **Figure 4** and have removed the x-axes in **Figure 5**, instead including scale bars with units to match the style of **Figure 4**.

Reviewer #3

The authors have fully addressed my previous review comments. I only have very minor suggestion for further improvement on their revision.

Fig S7. It would be more informative to use phylogram (branch length in the unit of genetic distance) instead of cladogram. Also, clearer if the length of sequence, unit of the tree branch, which genes are used to build the tree is indicated in the figure legend.

Same comments on Figs 4,5 and Fig S5.

Tommy Lam

Thank you for your feedback. We have reproduced all four trees as phylograms and updated the legends or the scale bars with the requested information.